# Towards a method to anticipate dark matter signals with deep learning at the LHC

**Ernesto Arganda**[1,2]⋆, **Anibal D. Medina**[2]†,
**Andres D. Perez**[2]‡ **and Alejandro Szynkman**[2]§

**1** Instituto de Física Teórica UAM/CSIC, C/ Nicolás Cabrera 13-15,
Campus de Cantoblanco, 28049, Madrid, Spain
**2** IFLP, CONICET - Dpto. de Física, Universidad Nacional de La Plata,
C.C. 67, 1900 La Plata, Argentina

⋆ ernesto.arganda@csic.es, † anibal.medina@fisica.unlp.edu.ar,
‡ andres.perez@iflp.unlp.edu.ar, § szynkman@fisica.unlp.edu.ar

## Abstract

We study several simplified dark matter (DM) models and their signatures at the LHC using neural networks. We focus on the usual monojet plus missing transverse energy channel, but to train the algorithms we organize the data in 2D histograms instead of event-by-event arrays. This results in a large performance boost to distinguish between standard model (SM) only and SM plus new physics signals. We use the kinematic monojet features as input data which allow us to describe families of models with a single data sample. We found that the neural network performance does not depend on the simulated number of background events if they are presented as a function of $S/\sqrt{B}$, for reasonably large $B$, where $S$ and $B$ are the number of signal and background events per histogram, respectively. This provides flexibility to the method, since testing a particular model in that case only requires knowing the new physics monojet cross section. Furthermore, we also discuss the network performance under incorrect assumptions about the true DM nature. Finally, we propose multimodel classifiers to search and identify new signals in a more general way, for the next LHC run.



# 1 Introduction

The standard model of particles (SM) provides a remarkably successful description of the elementary particle phenomena explored so far without precedent. After the Higgs boson discovery at the Large Hadron Collider (LHC) [1, 2], the last particle predicted by the theory undetected until then, the SM is complete. However, there are observations that point towards physics beyond the SM. One of the most intriguing enigmas in modern science is the presence of dark matter (DM) in the Universe revealed by cosmological observations [3–5]. The SM provides no viable DM candidate, and despite vast efforts in both experimental and theoretical aspects, its nature remains elusive.

In this work we focus on the usual monojet plus missing transverse energy channel to discriminate DM signatures at the LHC. To be as general as possible, we consider some simplified models to represent several possible DM frameworks. Some examples of DM collider searches with simplified models, including monojet analysis, can be found in Refs. [6–19]. We study

Axion-Like Particles (ALP) as DM, and three theoretical models with a mediator (with spin-0, 1 and 2) and a DM candidate.

In this context, the rise of machine learning (ML) techniques applied to high energy physics in recent years [20–22] provides a powerful tool to analyze an ever increasing amount of data[1]. Even more, new algorithms developed recently, like deep neural networks (DNN), can handle subtle signals that would be very hard to disentangle from the SM background with conventional methods [24]. These new techniques are increasingly being applied to hunt DM signatures in collider searches [25, 26], direct detection experiments [27], and cosmological probes [28–30].

We employ DNN with a supervised approach that would allow us to obtain good performances, but implies the construction of labeled data samples for every scenario considered. To tackle this issue, we use the kinematic monojet features as input data and study the distributions of our models. Since the coupling values and the type of DM candidate do not modify the kinematic distributions, a family of models can be described with the same data sample. Furthermore, the frameworks with a mediator have two other free parameters, the mediator and DM masses. We will see that only one of the dark sector masses modifies the kinematic variables, if we divide the parameter space according to the mediator status (on-shell or off-shell). These facts decrease the number of data samples needed to accurately represent a DM scenario in a general way.

To train the algorithms we organize the data in 2D histograms instead of event-by-event arrays. The joint distribution of both monojet transverse momentum and pseudorapidity provides additional information, and results in a large performance boost to distinguish between SM-only and SM plus new physics signals. The histogram representation has been proposed and used in a recent work [26] to discriminate among DM models in colliders with a signal-only hypothesis, and in Ref. [31] to distinguishing $W'$ signals. In this work, we include the SM background when new physics samples are prepared. Furthermore, we show important DNN properties that simplify and generalize even more the search for DM. The DNN results do not depend on the simulated number of SM events within the explored range when we present the performance as a function of $S/\sqrt{B}$, where $S$ and $B$ are the number of signal and background events per histogram, respectively. Additionally, we will see that the DNN is flexible enough to handle small variations in $S/\sqrt{B}$, in the masses of the dark sector, and even if an incorrect framework is applied as long as the kinematic distributions remain similar.

To ease the blind DM identification and discrimination from the SM background, we explore multiclass classifiers, i.e. DNN trained with more than one DM scenario. This approach shows good efficiencies, and even slightly outperforms some individually trained DNNs. Two methods are explored, a binary classifier prepared to distinguish SM-only histograms from samples with SM plus new physic events generated from several benchmark models, and a multiclass classifier to identify the most likely DM scenario among those considered. In the latter, crucial information about the kinematic distributions and therefore hints of the true underlying model can be extracted.

It is important to notice that the histogram approach tries to determine if there is new physics within a set of events. Unlike the event-by-event method, it does not classify each individual event. Therefore, we propose that the DNNs with histograms could be used as an initial analysis. Their performance turns out to be independent of the simulated number of background events, thus we can check if a set of events contains new physics with a relatively fast and general approach. If the network points towards a positive result, other analysis guided by the information provided by the histogram method should be applied to establish

---

[1]For a recent repository-review of ML techniques applied to particle physics with the latest works and developments regularly updated see [23].

the significance of the candidate signal[2].

We organize the paper as follows. In Section 2, we present the characteristics of the simplified DM models we are testing, and the sample generation employed. Since the kinematic distributions are the input data in our ML algorithms, their relevant features are analyzed, and we define benchmark models to work with. Then, we describe the structure of our ML techniques in Section 3, compare DNN with convolutional neural networks (CNN) results, and also implement two data organizations: event-by-event analysis and 2D-histogram representation. In Section 4, we explore the flexibility of the method, showing that the DNN performance does not change significantly with the simulated number of background events. We also show the discrimination power among different DM frameworks, and study the impact on the results when incorrect assumptions are considered, i.e. discrepancies in $S/\sqrt{B}$ and in new physics models between train and test data samples. Finally, in Section 5 we propose multimodel classifiers to search and identify new signals in a more general way, compare their performances, and explore their responses testing new models for which they were not trained. A discussion about the results, strength and weakness of the different methods and approaches analyzed can be found in Section 6. The conclusions are left for Section 7.

## 2 Models and sample generation

In this section we describe four DM theoretical frameworks chosen in this work. We consider the dark sector couplings and masses as free parameters, hence each framework represents a family of DM models. We will see that for our collider study, the masses of the dark sector determine the signal kinematic distribution. On the other hand the coupling values, along with the type of DM, are only involved in the cross section of the relevant processes.

To be as general as possible, we discuss the kinematic features which are going to be used as input data for the ML algorithms to disentangle DM scenarios from the SM-only hypothesis. Finally, we select a few benchmark models to be used in next sections.

### 2.1 Simplified models

We study ML techniques applied to the supervised classification of samples from the following four DM theoretical frameworks:

- DM with a spin-0 mediator,

- DM with a spin-1 mediator,

- DM with a spin-2 mediator,

- Axion-Like Particle as DM.

**DM with a spin-0 mediator**

We consider a simplified model where the interaction of a DM candidate with the SM is mediated by a spin-0 field, $Y_0$, dubbed mediator. Three types of DM candidates, $X$, are taken into account: a Dirac fermion $X_D$, a real scalar $X_R$, and a complex scalar $X_C$. The renormalizable interactions of the mediator with the DM can be described by the following Lagrangian [32–34]

---

[2]Our main focus is to asses the viability of the histogram approach. In that sense, determining signal significances of this histogram method is not straightforward and requires a more in-depth study that is beyond the scope of this work.

$$L_{DM}^{Y_0} = \frac{1}{2} g_{X_R}^S M_{X_R} X_R X_R Y_0 + g_{X_C}^S M_{X_C} X_C^* X_C Y_0 + \bar{X}_D (g_{X_D}^S + i g_{X_D}^P \gamma_5) X_D Y_0 \,, \tag{1}$$

where quadratic terms in the mediator field have been omitted (the Majorana case can be easily obtained from the Dirac scenario). In order to take dimensionless couplings we include DM masses as natural scales in dimension-3 operators. These factors can be readjusted modifying the coupling strengths if a different normalization scale is needed. The renormalizable interactions of the mediator field with SM quarks can be written as

$$L_{SM}^{Y_0} = \sum_{i,j} \left( \bar{d}_i \frac{y_{ij}^d}{\sqrt{2}} (g_{d_{ij}}^S + i g_{d_{ij}}^P \gamma_5) d_j + \bar{u}_i \frac{y_{ij}^u}{\sqrt{2}} (g_{u_{ij}}^S + i g_{u_{ij}}^P \gamma_5) u_j \right) Y_0 \,, \tag{2}$$

where $d$ and $u$ denote down- and up-type quarks, respectively, $i,j = 1,2,3$ are flavor indices, and $g^{S/P}$ are the scalar/pseudo-scalar couplings of DM and quarks, which we take diagonal to avoid large flavor-changing neutral current interactions. These couplings are normalized to the SM Yukawa couplings, $y_{ii}^f = \sqrt{2} m_f / v$, assuming an ultraviolet complete description of the theory with the couplings of the messenger to the SM particles proportional to the particle masses. This implies that interactions with light quarks are strongly suppressed, hence the main channels for DM production in this model involve top-quark loops.

Higgs portal models are included in this framework. Also, the possibility that the SM Higgs itself takes the role of the spin-0 mediator $Y_0$. However, as we will see in the following sections, the latter scenario can not be discriminated from the SM background with the methods proposed in this work if the SM Higgs is produced on-shell (see Appendix C).

**DM with a spin-1 mediator**

In this case, we consider a simplified model with a DM candidate and a spin-1 mediator, $Y_1$. The DM-messenger interactions can be described by the following renormalizable Lagrangian [32–34]

$$L_{DM}^{Y_1} = \frac{i}{2} g_{X_C}^V \left[ X_C^* (\partial_\mu X_C) - (\partial_\mu X_C^*) X_C \right] Y_1^\mu + \bar{X}_D \gamma_\mu (g_{X_D}^V + g_{X_D}^A \gamma_5) X_D Y_1^\mu \,. \tag{3}$$

Then, interactions between the mediator and SM quarks can be expressed as

$$L_{SM}^{Y_1} = \sum_{i,j} \left( \bar{d}_i \gamma_\mu (g_{d_{ij}}^V + g_{d_{ij}}^A \gamma_5) d_j + \bar{u}_i \gamma_\mu (g_{u_{ij}}^V + g_{u_{ij}}^A \gamma_5) u_j \right) Y_1^\mu \,, \tag{4}$$

where $g^{V/A}$ are the vector/axial-vector couplings of DM and quarks, and, as in the spin-0 case, are also considered diagonal to avoid large flavor-changing neutral current interactions.

This framework includes the usual supersymmetric (SUSY) neutralino, with the rest of the SUSY spectrum sufficiently heavy to be decoupled. A weakly-interacting massive particle (WIMP) like benchmark model will be considered during the rest of this work, however notice that the couplings are not specified. The goal is to include not only SUSY models but to keep a broad approach.

**DM with a spin-2 mediator**

We consider DM particles which interact with SM particles via a spin-2 mediator, $Y_2$. The DM-messenger interaction Lagrangian of this framework is given by [35]

$$L_{DM}^{Y_2} = -\frac{1}{\Lambda} g_X^T T_{\mu\nu}^X Y_2^{\mu\nu} \,, \tag{5}$$

where $\Lambda$ is the scale at which new physics enters, $g_X^T$ a dimensionless coupling parameter, and $T_{\mu\nu}^X$ the energy-momentum tensor of the DM field that can be expressed as

$$T_{\mu\nu}^{X_R} = -\frac{1}{2}g_{\mu\nu}(\partial_\rho X_R \partial^\rho X_R - m_{X_R}^2 X_R^2) + \partial_\mu X_R \partial_\nu X_R, \tag{6}$$

$$T_{\mu\nu}^{X_D} = -g_{\mu\nu}(\bar{X}_D i\gamma_\rho \partial^\rho X_D - m_{X_D}\bar{X}_D X_D) + \frac{1}{2}g_{\mu\nu}\partial_\rho(\bar{X}_D i\gamma^\rho X_D)$$
$$+ \frac{1}{2}\bar{X}_D i(\gamma_\mu \partial_\nu + \gamma_\nu \partial_\mu)X_D - \frac{1}{4}\partial_\mu(\bar{X}_D i\gamma_\nu X_D) - \frac{1}{4}\partial_\nu(\bar{X}_D i\gamma_\mu X_D), \tag{7}$$

$$T_{\mu\nu}^{X_V} = -g_{\mu\nu}\left(-\frac{1}{4}F_{\rho\sigma}F^{\rho\sigma} + \frac{1}{2}m_{X_V}^2 X_{V\rho}X_V^\rho\right) + F_{\mu\rho}F_\nu^\rho + m_{X_V}^2 X_{V\mu}X_{V\nu}, \tag{8}$$

with $F_{\mu\nu}$ the field strength tensor.

Interactions between the mediator and SM particles are obtained by

$$L_{SM}^{Y_2} = -\frac{1}{\Lambda}\sum_i g_i^T T_{\mu\nu}^i Y_2^{\mu\nu}, \tag{9}$$

where $i$ denotes the SM fields (Higgs doublets, quarks, leptons, and gauge bosons). The corresponding energy-momentum tensors can be found in Ref. [36]. When a universal coupling between the spin-2 mediator and the SM particles is assumed, we get the original Randall-Sundrum (RS) model of localized gravity [37].

**Axion-Like Particle as DM**

In this model, the SM is extended with the inclusion of a CP-odd scalar field, singlet under the SM gauge symmetries. This new field is assumed to be the (pseudo) Nambu-Goldstone boson of a spontaneously broken $U(1)$ symmetry, associated with a heavy scale $f_a$, i.e. an ALP, $a$. Its couplings are purely derivative and weighted by inverse powers of $f_a$, then we consider that the ALP mass is much lighter than this scale. The effective Lagrangian involving ALP in linear realization can be written as [38]

$$L_a = \frac{1}{2}(\partial_\mu a)(\partial^\mu a) - g_W \frac{a}{f_a}W_{\mu\nu}^a\widetilde{W}^{a\mu\nu} - g_B \frac{a}{f_a}B_{\mu\nu}\widetilde{B}^{\mu\nu} - g_G \frac{a}{f_a}G_{\mu\nu}^a\widetilde{G}^{a\mu\nu}$$
$$+ \left[ig_a \frac{a}{f_a}(\bar{Q}_L y^u \widetilde{H}u_R - \bar{Q}_L y^d H d_R - \bar{L}_L y^e H e_R) + h.c.\right], \tag{10}$$

where $\widetilde{X}^{\mu\nu} = 1/2\epsilon^{\mu\nu\rho\sigma}X_{\rho\sigma}$ are the field strength duals. For the monojet channel, the most important ALP coupling is to gluons.

## 2.2 Sample generation

We use `MadGraph5_aMC@NLO v2.7.3` [39] to generate events with monojets plus missing energy at parton level. Then, parton shower and hadronization are performed with `Pythia 8.2.44` [40], and the detector-level data is simulated using `Delphes 3.4.2` [41] with the default ATLAS card. The processes are generated using the `FeynRules` [42,43] model database. For our analysis on DM collider searches, we focus on the monojet and missing transverse energy channel[3]. We generate the following processes for the theoretical frameworks with a mediator

$$pp \to \mathrm{DM}\,\mathrm{DM}\,j, \tag{11}$$

---

[3]In Ref. [26] it is shown that the inclusion of an additional hard jet in the analysis does not result in a significant change, with a study of similar characteristics to the one used in this work and the same basic cuts.

and for the ALP model

$$pp \rightarrow aj. \tag{12}$$

A simulated center-of-mass energy of $\sqrt{s} = 14$ TeV is used, and generation level cuts of $p_T^j \geq 130$ GeV and $|\eta^j| \leq 5$ are imposed for the leading jet. For each model analyzed, 0.5M events are produced.

We would like to mention that frameworks involving a messenger have two key free parameters: the mediator mass, $m_Y$, and the DM mass, $m_{DM}$. Although the type of DM candidate and coupling values are involved in the total number of new physics events, we have checked that for a given framework these variables do not modify the monojet kinematic distributions. In particular, we use a Dirac fermion and couplings equal to 1 in our simulation. As we will see in the next section, each choice of $(m_Y, m_{DM})$ determines a model with distinctive characteristics that could lead to different collider signals. Therefore, several sets of 0.5M events are generated per framework. On the other hand, in ALP models we consider a very light $a$, which behaves as a massless particle for typical LHC energies. We have checked that variations in $m_a$ (ALP mass) and $f_a$ produce the same collider signatures. Then, only one set of events is generated for this framework, with a very light ALP mass, $m_a = 10^{-5}$ GeV and $f_a = 10^4$ GeV. Monojet processes for every framework can be found in Appendix A.

Finally, for the monojet background we consider the following process

$$pp \rightarrow Zj(Z \rightarrow \nu\bar{\nu}). \tag{13}$$

An extended discussion about the dominant SM-background channel, and the role of other subdominant contributions to the monojet signal can be found in Appendix B. For the SM model, 1.5M events are generated with the same center-of-mass energy and generation cuts used in the new physics cases.

## 2.3 Kinematic distributions

In Fig. 1 we show the monojet kinematic distributions for some examples of the frameworks considered in this work, at parton level. For this particular choice of $(m_Y, m_{DM})$, the simplified frameworks with a mediator are called *on-shell* for all spin values. This notation refers to the mediator status in the monojet channel which is produced resonantly. Considering different choices of $(m_Y, m_{DM})$ distinct data sets have to be simulated and we obtain various kinematic distributions. On the other hand, for the ALP and SM background cases only one data sample was generated since just one scenario describes the model behavior, as we discussed in the previous section. We would like to highlight that we present the fraction of events, the total number of events that would be produced at the LHC depends on the luminosity and the couplings of each model, but those variables do not modify the kinematic distributions.

We can see that the examples with a mediator present a harder $p_T^j$ distribution than the ALP. Even more, the latter model and the SM describe very similar distributions, especially at low $p_T^j$ where the higher fraction of events is found. Therefore, ALP should be harder to discriminate from the background only hypothesis.

All the models present a pseudorapidity distribution similar to the SM, except for the spin-1 mediator case. This feature provides complementary information, as can be seen comparing the examples with spin-1 and spin-2 mediators. The latter model presents a harder $p_T^j$ distribution, but a $\eta^j$ distribution similar to the SM, unlike the spin-1 case.

Finally, the flat jet azimuthal angle distribution does not show any useful structure to distinguish any model from the SM background or each other, as expected by the symmetry of the processes. This will allow us to dismiss $\phi^j$, and to arrange $p_T^j$ and $\eta^j$ information into 2D histograms for the ML algorithms proposed.

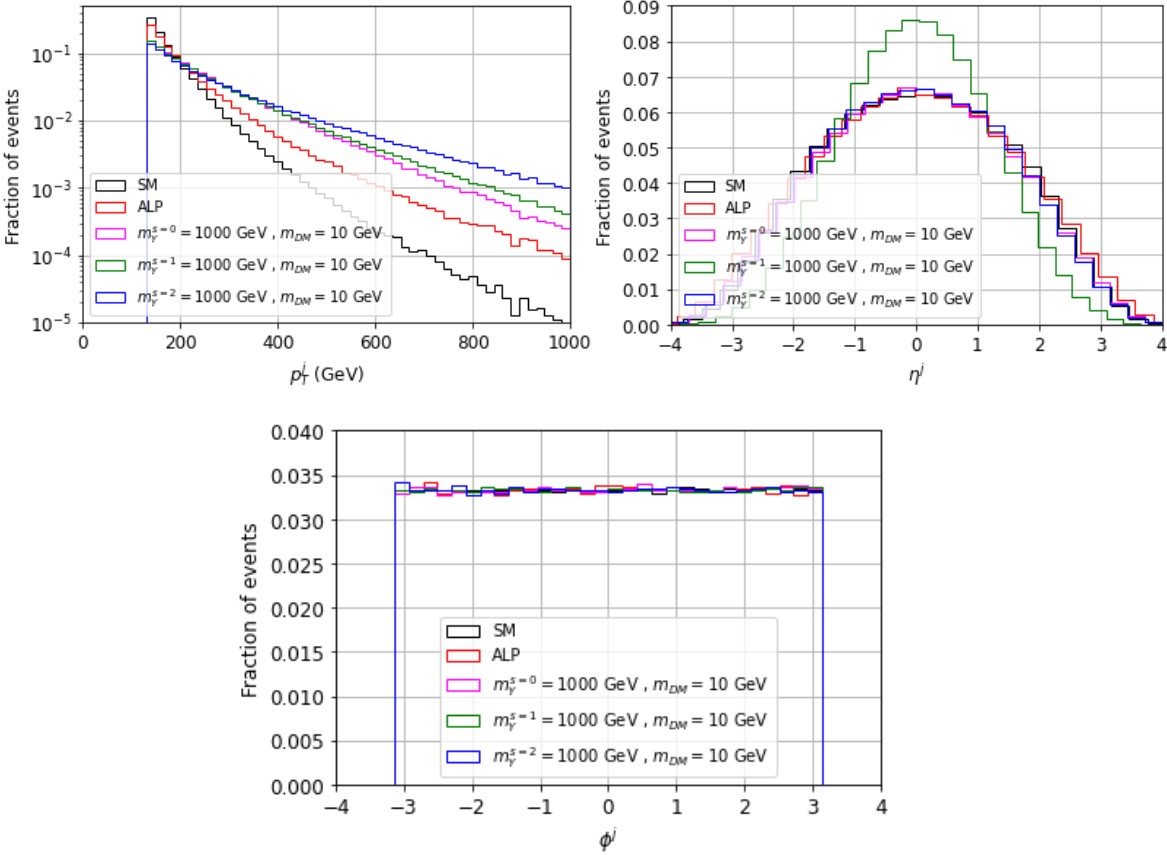

Figure 1: Kinematic distribution for the frameworks considered in this work at parton-level. For models with a mediator, these examples belong to the *on-shell* regime.

Next, we describe the impact of different $(m_Y, m_{DM})$ values in frameworks with a mediator. For simplicity, we only show the $p_T^j$ kinematic distributions for the spin-1 case. Spin-0 and spin-2 frameworks present a similar behavior. In Fig. 2, we study the $p_T^j$ kinematic distribution of three regimes, defined according to the status of the mediator: *off-shell* (top-left panel), *on-shell* (top-right panel), and *off-shell by phase space* or *off-shell PS* (bottom panel), where the latter regime refers to a scenario with $m_Y > 2m_{DM}$, but the mediator can not be produced on-shell due to the energy of the collision. We can see that in each regime, the kinematic distribution does not depend on one of the dark sector masses. Since the coupling values only determine the total number of events but do not modify the kinematic distributions, every model is defined by a single parameter, $m_{DM}$ or $m_Y$.

Hence, in Fig. 3 we show, for the spin-1 case, the hardest and softest distributions that can be found in each regime. On the top-left panel, the *off-shell* case, the distribution tends to the SM one for $m_{DM} \to m_Y/2$ and the hardest one is obtained for $m_{DM} \sim 2000$ GeV (higher values result in a softer distribution). On the top-right panel, the *on-shell* regime, the distribution is similar to the SM background for $m_Y \lesssim m_Z$, and reaches its maximum for $m_Y \sim 3000$ GeV. On the bottom panel, *off-shell PS* case, the distribution is only slightly modified with $m_{DM}$ between $\sim 100$-$1000$ GeV, and never tends to the SM one. The distributions for the spin-0 and spin-2 frameworks can be seen in Appendix C, for the same three regimes. Notice there that the only framework where there is no $(m_Y, m_{DM})$ value whose distribution is similar to the SM

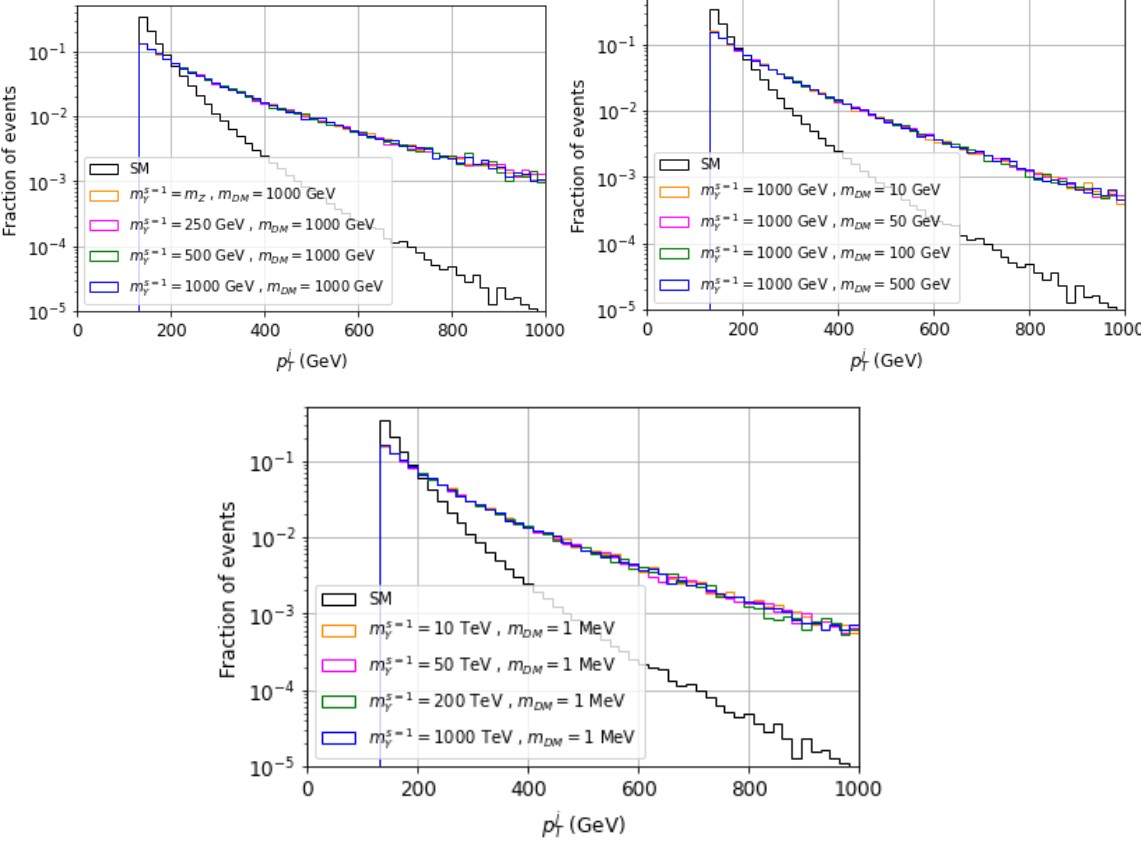

Figure 2: Spin-1 mediator framework, parton-level data. The top-left, top-right, and bottom panels show that the $p_T^j$ distribution for the *off-shell*, *on-shell*, and *off-shell PS* regime does not depend on $m_Y$, $m_{DM}$, and $m_Y$, respectively.

background is the spin-2 case.

Notice that some curves within the same frameworks but with different values of ($m_Y$, $m_{DM}$) overlap, or are extremely similar. This is the case for the examples shown in Fig. 2. Also, some cases shown in Fig. 3 from different regimes are very similar to each other, e.g. ($m_Y$, $m_{DM}$) = ($m_Z$, 2000 GeV), (3 TeV, 10 GeV), and (10 TeV, 1000 GeV), the corresponding blue curves on the top-left (*off-shell*), top-right (*on-shell*), and bottom (*off-shell PS*) panels. Even more, as can be seen in Appendix C, curves between different frameworks can overlap.

Since we use the kinematic information as input data for our algorithms, this degeneracy can not be disentangle by the ML methods in many cases. Instead of discriminating between particular models, the algorithm classifies families of models with similar kinematic distributions.

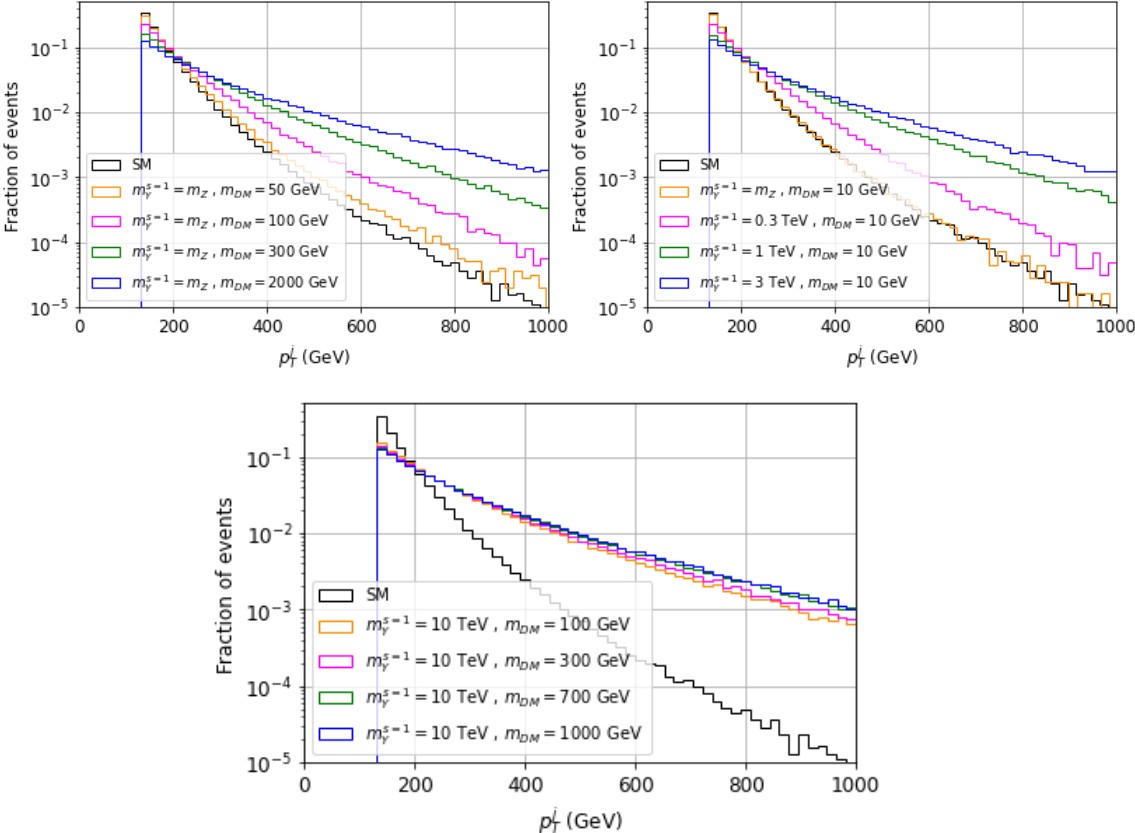

Figure 3: $p_T^j$ distributions of the spin-1 mediator framework, *off-shell* (top-left panel), *on-shell* (top-right panel), and *off-shell PS* (bottom panel) regimes, at parton level.

### 2.3.1 Benchmark models

To study the performance of the ML algorithms used in this work, we define ten benchmark models shown in Table 1. We consider ALPs and for each framework with a mediator we select three models, one per regime: *off-shell*, *on-shell*, and *off-shell PS*. In Section 3 we will test the benchmark models against SM-only samples individually. On the other hand, in Section 5 we will study methods trained with several benchmark models, called multimodel. To evaluate their performance under new unexpected data, three benchmark models are not used in the training set and therefore do not have a label assigned.

In Fig. 4, we show the $p_T^j$ and $\eta^j$ distributions corresponding to the mentioned benchmark models, for the *off-shell* (top row), *on-shell* (middle), and *off-shell PS* (bottom row) regimes. We include the SM background and ALP distributions in each panel as reference to guide the eye. Unlike in the previous section, we employ detector-level data. Notice that no significant differences can be seen and the overall behavior is the same between parton and detector-level representations. Nonetheless, we will use detector-level simulations to evaluate the ML algorithms performance.

Table 1: Benchmark models and labels used in individual binary, and in multimodel binary and multiclass classifiers. Models without a label are not used to train the corresponding classifier.

| Benchmark Model | | Label | | |
| --- | --- | --- | --- | --- |
| | | Individual | Multimodel | |
| | | Binary | Binary | Multiclass |
| | SM | 0 | 0 | 0 |
| | ALPs | 1 | 1 | 1 |
| Spin-0 mediator | $m_Y^{s=0} = m_Z$ , $m_{DM} = 300$ GeV | 1 | 1 | 2 |
| | $m_Y^{s=0} = 1000$ GeV , $m_{DM} = 10$ GeV | 1 | 1 | 3 |
| | $m_Y^{s=0} = 10$ TeV , $m_{DM} = 10$ GeV | 1 | - | - |
| Spin-1 mediator | $m_Y^{s=1} = m_Z$ , $m_{DM} = 300$ GeV | 1 | 1 | 4 |
| | $m_Y^{s=1} = 1000$ GeV , $m_{DM} = 10$ GeV | 1 | - | - |
| | $m_Y^{s=1} = 10$ TeV , $m_{DM} = 10$ GeV | 1 | 1 | 5 |
| Spin-2 mediator | $m_Y^{s=2} = m_Z$ , $m_{DM} = 300$ GeV | 1 | - | - |
| | $m_Y^{s=2} = 1000$ GeV , $m_{DM} = 10$ GeV | 1 | 1 | 6 |
| | $m_Y^{s=2} = 10$ TeV , $m_{DM} = 10$ GeV | 1 | 1 | 7 |

# 3 Machine Learning algorithms

In this section we describe the structure of our supervised ML techniques. All the algorithms are constructed in `Python 3.7.6` [44] using the library `Keras` [45] along with `TensorFlow` [46] for backend implementation. We will compare several ML methods, and also two different data organizations. Although we study multiple benchmark models, below we will test each new physics model versus SM background individually, with a binary classifier.

## 3.1 Neural networks with event-by-event data

To compare the performance of different techniques, as a first step we use deep neural networks (DNNs) as individual binary classifiers for the monojet data at detector level in the form of arrays, i.e. the networks take as input data event-by-event information. In this setup the input parameters are the three kinematic features of the monojet, $p_T^j$, $\eta^j$, $\phi^j$. We consider a sample size of 0.5M events for every benchmark model, as well as for the SM background. Data samples are divided with a 0.64:0.16:0.20 train-validation-test ratio. To avoid over-training we include dropout in every hidden layer and call for an early stopping if the validation loss has not improved for over 100 epochs. More details of the DNN structure can be found in Table 2.

We apply this algorithm to the ten benchmark models defined in Section 2.3.1 (see Table 1). We would like to highlight that in individual binary classifiers, each new physics benchmark model (events labeled as '1') is tested against the SM background (events labeled as '0') independently. Hence we analyze ten DNN algorithms whose results are presented in Fig. 5, and we include the ALP benchmark in every panel for reference. We show the Receiver Operating Characteristic (ROC) curves, and as a measure of the DNN performance we calculate the areas under the ROC curves (AUCs), a conventional metric to test binary classifiers (AUC=1 is a perfect classifier, and AUC=0.5 a random classifier).

Figure 4: $p_T^j$ and $\eta^j$ distribution for the benchmark models shown in Table 1. *Off-shell* (top row), *on-shell* (middle row), and *off-shell PS* (bottom row) regimes, at detector level.

As expected, the discrimination power depends strongly on the model[4]. With this technique we obtain AUC $\simeq 0.57$ for the ALP framework, which means that it would not be possible to identify a DM signal. With respect to the benchmark models with a mediator we get AUC $\simeq 0.70 - 0.78$, hence they can be discriminated from the SM background with a rather low efficiency. In the next section we will study a better performing method.

The general behavior of the previous result can be seen from the input features kinematic

---

[4]We have found that increasing the jet transverse momentum lower bound, $p_T^{j-min}$, decreases the network performance for all benchmark models, except in the ALP case for which a small improvement is observed ($\lesssim 5\%$ for $p_T^{j-min}$ within $130-400$ GeV). We have also checked that modifying the pseudorapidity upper bound, $|\eta^j| \in [2,5]$, do not change our results.

Table 2: ML algorithms specifications. For the CNN case, we denote $(n \times m, l)$, or $(n \times m)$, a layer with a $n \times m$ kernel dimension, and $l$ nodes in each convolutional layer.

| | DNN | CNN |
|---|---|---|
| **Input data** | | |
| Format | event-by-event | 2D histogram |
| Features | $(p_T^j, \eta^j, \phi^j)$ | $p_T^j$ vs $\eta^j$ |
| Sample size | 0.5M per model | 20k histograms per model |
| Train:Validation:Test | | 0.64:0.16:0.20 |
| **Neural network** | | |
| Layers | 3 dense layers | Convolutional 2D layer (3×3, 32) |
| | Hidden layers nodes = 20 | Max pooling 2D layer (2×2) |
| | Dropout in every hidden layer: 0.2 | Convolutional 2D layer (3×3, 64) |
| | | Max pooling 2D layer (2×2) |
| | | Flatten layer |
| | | Hidden dense layer (20 nodes) |
| Activation function | | Hidden layers: *relu* |
| | | Output layer: *sigmoid* |
| **Compilation** | | |
| Loss function | | *binary_crossentropy* |
| Optimizer | | *adam* (initial learning rate = 0.0001) |
| Metric | | *accuracy* |
| Batch size | | 128 |
| Max epochs | | 1500-2500 |
| Patience | | 100-300 epochs |

distributions shown in Fig. 4. The new physics models present a higher fraction of events than the $Z$+jet SM background for higher values of $p_T^j$ and values of $\eta^j$ closer to zero. Furthermore, models with higher discrepancies with respect to the SM result in higher AUCs when are applied to ML classification algorithms. This trend will be present throughout the rest of our work.

## 3.2   Neural networks with data as histograms

As discussed previously, the jet azimuthal angle $\phi^j$ does not provide any information to distinguish new physics signals from SM processes. Therefore, we can construct 2D histograms made from the pair $(p_T^j, \eta^j)$, which can provide additional information on the joint distribution for both kinematic observables. This method has been proposed and used in Ref. [26] to discriminate among monojet DM signatures in colliders with a signal-only hypothesis, and in Ref. [31] to distinguishing $W'$ signals. In this work, we include the SM background when new physics samples are prepared in the DM search.

We impose the following selection cuts: $130 \le p_T^j \le 2000$ GeV, and $|\eta^j| \le 4$. The parameter space is arranged into 30×30 bins, dividing each range linearly. Then, we construct two sets of 20k 2D-histogram samples per hypothesis[5]. The first contains a combination of

---

[5]For our simulated events, we have checked that the network performance saturates for data sets with more

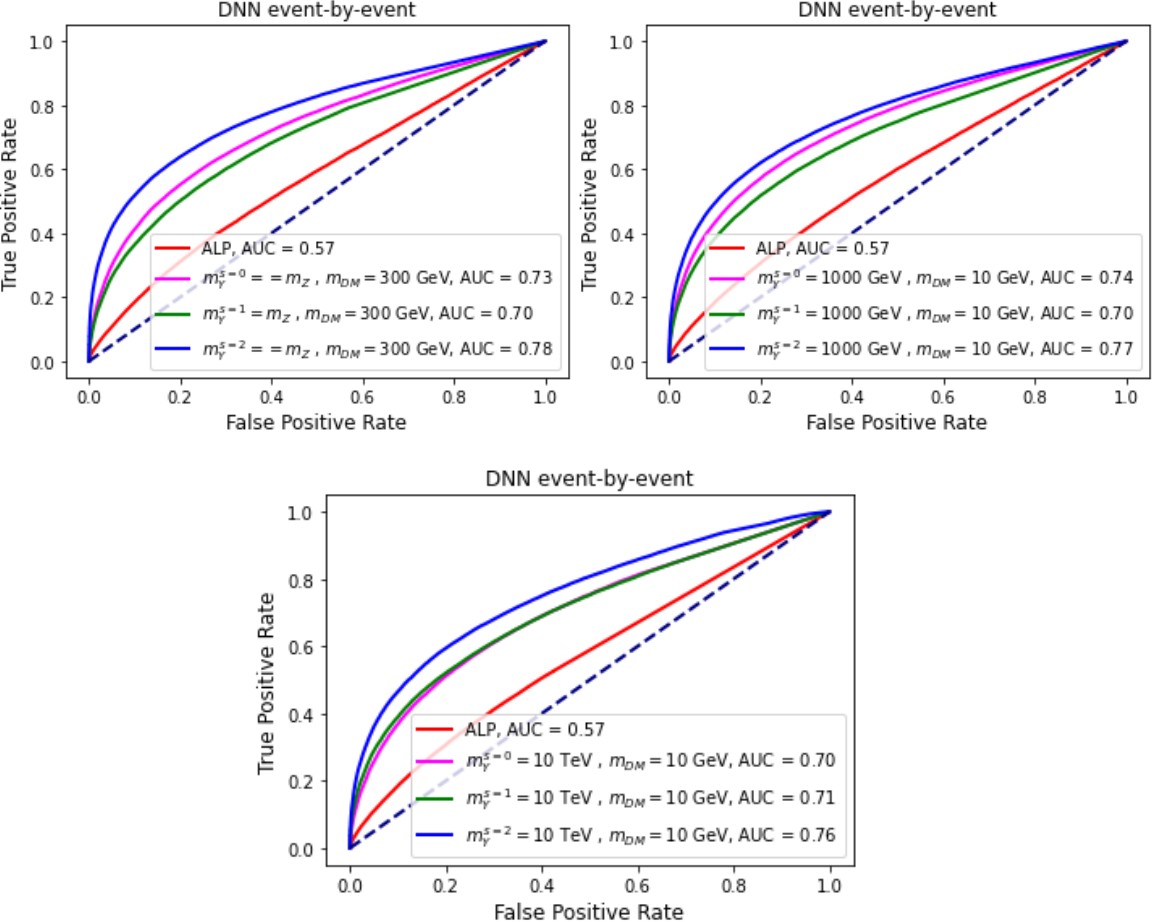

Figure 5: ROC curves for new physics benchmark models vs SM background, using DNN and event-by-event data.

$S$ simulated new physics events and $B$ simulated SM events, the other set is generated only with SM events. Notice that the DNN considers each histogram as one sample, not individual events. Therefore, to avoid including bias during the training process we use the same amount of histograms labeled '1' (signal, i.e. generated with SM plus new physics events) and '0' (no signal, i.e. with SM-only events). To build each histogram we choose the events at random over the total number of simulated processes. In this way, an event can be in more than one histogram, but obviously a histogram does not contain the same event more than once[6].

To use a DNN, we normalize the events per bin. Each 2D histogram, i.e. a 30×30 matrix with the number of events per bin as elements, is divided by the value of the bin with maximum amount of events. For CNN, the matrices are used as input data. But to be used as DNN input data, each matrix is unrolled into a 1D array. We would like to remark here that each 2D histogram (or array), generated with multiple individual events, is a single training example.

In Fig. 6 we show several 2D-histogram examples considering the simulated events at detector level. Each row presents figures generated with only SM data or with one benchmark model plus SM events. Every plot mixing signal and background was constructed with $B = 50$k SM-background events, and $S$ new physics events with the ratio $S/B = 0.1, 0.01, 0.001$ as

---

than 5k histograms. To be conservative, we have considered sets with 20k histograms.

[6]We employ this data augmentation technique to generate the needed sample size.

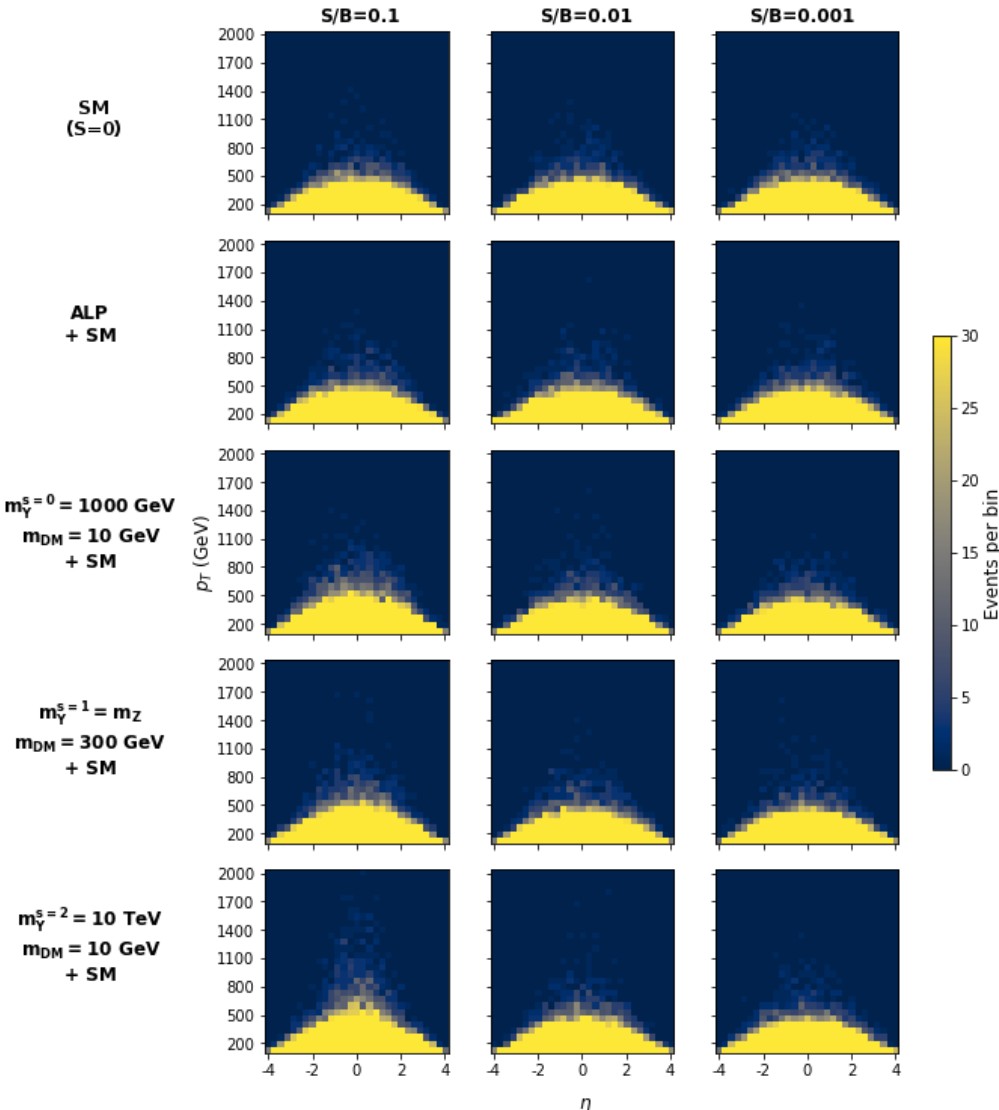

Figure 6: 2D histograms of $p_T^j$ vs $\eta^j$ with 30×30 bins for the monojet final state considering only SM events, and several benchmark models plus SM background. Each plot mixing signal and background was constructed with $B = 50k$ SM events and $S$ new physics events, at detector level. SM-only histograms have the same total number of events as their corresponding mixing plots. As an example, three $S/B$ ratios are shown. The color coding represents the number of events per bin, however the color scheme saturates at 30 events, to ease visualization.

indicated at the top of each column. SM-only histograms have the same total number of events as their counterparts mixing signal and background, to be able to compare both sets. The color coding represents the number of events per bin (not normalized to 1). However, the color scheme saturates at 30 events to ease visualization in regions with high $p_T^j$. We can see that if a histogram is constructed with decreasing values of new physics events $S$ (or decreasing values of $S/B$), it is increasingly difficult to distinguish between a new physics plus SM image and the corresponding SM-only image, as expected.

As in Section 3.1, with event-by-event data organization, the discrimination power depends

on the kinematic features of the models. Models with $p_T^j$ and $\eta^j$ kinematic distributions similar to the SM ones, like ALPs, are harder to disentangle by eye, even for high $S/B$. On the other hand, models with similar but harder distribution with respect to the SM, like spin-0 and spin-1 mediators with this particular choice of $(m_Y, m_{DM})$, are easier to discriminate for high signal-to-background ratio. Finally, the model with spin-2 mediator presents more number of events with higher $p_T$ that could be identify, with significant error, by naked eye. We will see that ML results with data as histograms show the overall trend mentioned, but allow a more efficient discrimination, even for very low $S/B$ ratios.

### 3.2.1 DNNs with data as histograms

In this case we consider DNNs as individual binary classifiers, and use the 2D histograms constructed with monojet kinematic data unrolled as 1D arrays as inputs. With this representation, binary classifiers compare histograms generated with only SM-background data (labeled as '0') against histograms with new physics plus SM events (labeled as '1'). We highlight again that individual events are no longer input examples, but each histogram is a single sample.

Data sets are divided with a 0.64:0.16:0.20 train-validation-test ratio[7]. We consider a sample size of 20k histograms for every benchmark model plus SM, and 20k histograms with only SM background. To avoid over-training we include dropout in every hidden layer and call for an early stopping if the validation loss has not improved for over 100 epochs. For some cases with low $S/B$, early stopping is raised to 300 epochs, and max epochs to 2500 if needed, with no signals of over-fitting. More details of the DNN structure can be found in Table 2.

In Fig. 7, we show the results of the benchmark models (see Table 1). For each DNN we repeat the training procedure 10 times and remove the largest and smallest AUC values, then we calculate the AUC mean with its standard deviation as a performance metric[8]. We include the standard deviation as error bars for all points in the figure, but in most cases the error bars are small. Notice that the models with spin-2 mediator present the largest variations. We would like to remark that

- each model is tested independently, i.e. we assume one new physics scenario at a time,

- each point represents a new DNN (all with the same structure), trained and tested with a data set generated with a specific model, $S$, and $B$ values,

- the value of $B$ (SM events per histogram) depends on the collider luminosity.

- the value of $S$ (signal events per histogram) is given by the collider luminosity, the choice of a new physics framework, and the process cross section. The latter is fixed by the election of the coupling values.

For example on the top-left panel, the blue point at $S/\sqrt{B} \simeq 10$ represents the result of a DNN whose histograms are constructed with $B = 50k$ and $S = 2236$ ($m_Y^{s=2} = m_Z$, $m_{DM} = 300$ GeV) events; and the blue point at $S/\sqrt{B} \simeq 0.1$ represents the result of a different DNN trained and tested separately, whose histograms have $B = 50k$ and $S = 22$ ($m_Y^{s=2} = m_Z$, $m_{DM} = 300$ GeV) events. Therefore, each curve shown in Fig. 7 can be interpreted as a particular model for which different coupling values are being tested with a new independent DNN, for a fixed LHC luminosity.

---

[7]We obtain the same results for different train-validation-test ratios.

[8]Unless it is stated explicitly, we do not perform this procedure throughout the rest of this work because the standard deviation is significant only for points where the method begins to be inefficient ($0.5 <$ AUC $\lesssim 0.7$). In these regions with low, but not null, discrimination power it is more computationally expensive to obtain a stable value.



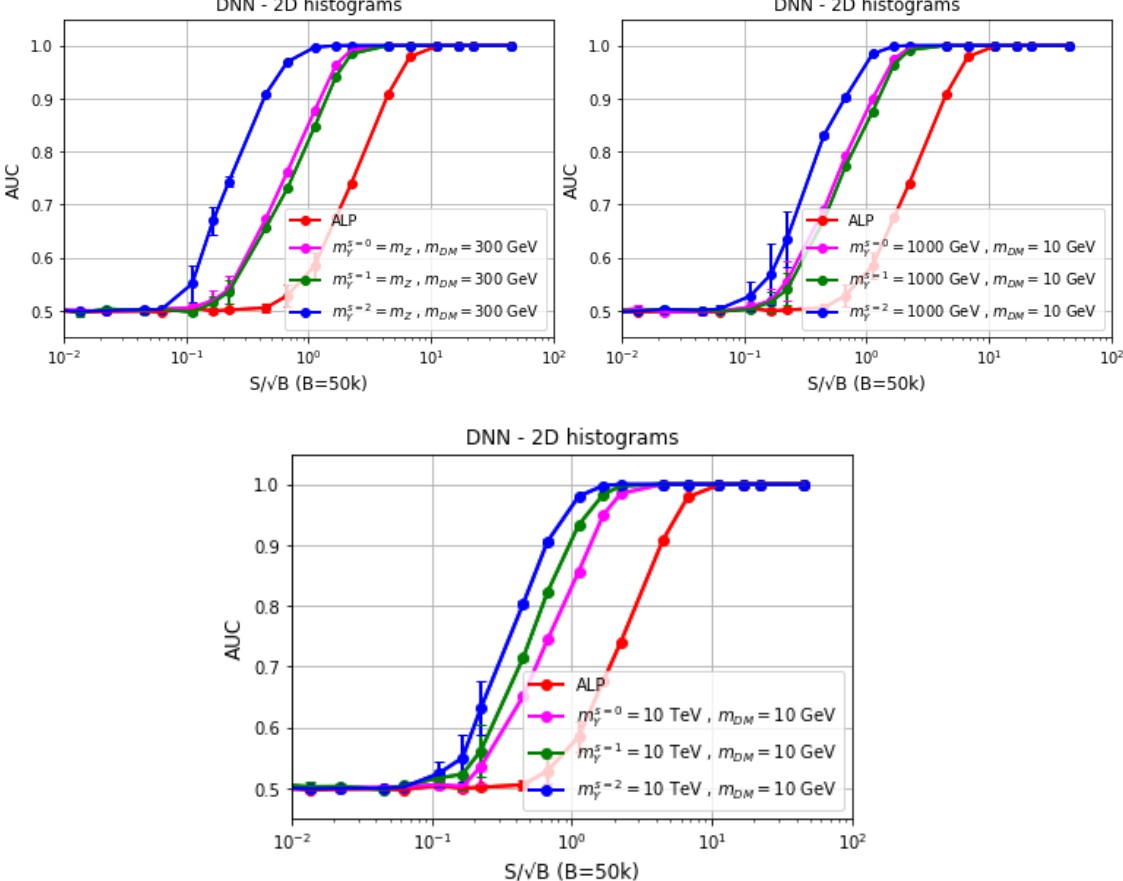

Figure 7: AUC results for several benchmark models, using DNN and 2D histograms as input data. Each model is tested independently in a binary classifier to distinguish new physics plus SM vs SM-only histograms. The training procedure was repeated 10 times and the largest and smallest AUC values where removed from each DNN to calculate the AUC mean with its standard deviation. Error bars are included for all points.

We can see that the AUC value monotonically increases for the $S/\sqrt{B}$ entire range, except for a few points around AUC = 0.5. Furthermore, the DNN algorithm could be a good classification tool for DM scenarios with very low signal-to-background ratios. For example, on the top-left panel we get AUCs > 0.9 for $S/\sqrt{B} \gtrsim 0.5, 1.2, 1.3, 4$ for spin-2, spin-0, spin-1 mediators, and ALP frameworks, respectively. Moreover, for $S/\sqrt{B} \gtrsim 1, 3, 4, 10$ respectively, we get AUCs $\sim 1$, i.e. almost a perfect classifier. It is important to notice that some of these values are smaller than the usual significance discovery threshold $\gtrsim 5$, however in this case $S/\sqrt{B}$ is a histogram parameter and we can not interpret it straightforwardly as a significance in an usual counting experiment. This kind of algorithm tries to determine whether or not there may be new physics in a histogram (constructed with $S$ new physics events and $B$ background events). Unlike in the event-by-event representation, the histogram method classifies sets of events, i.e. the DNN labels histograms as a whole, not each individual event.

To illustrate the results that would be obtained if data were analyzed with a DNN, we show in Fig. 8 the network outputs for three $S/\sqrt{B}$ choices, considering the benchmark model with $m_Y^{s=2} = 1000$ GeV, $m_{DM} = 10$ GeV (corresponding to the blue curve on the top-right panel of Fig. 7). The black curves represent the results when test samples of SM-only histograms

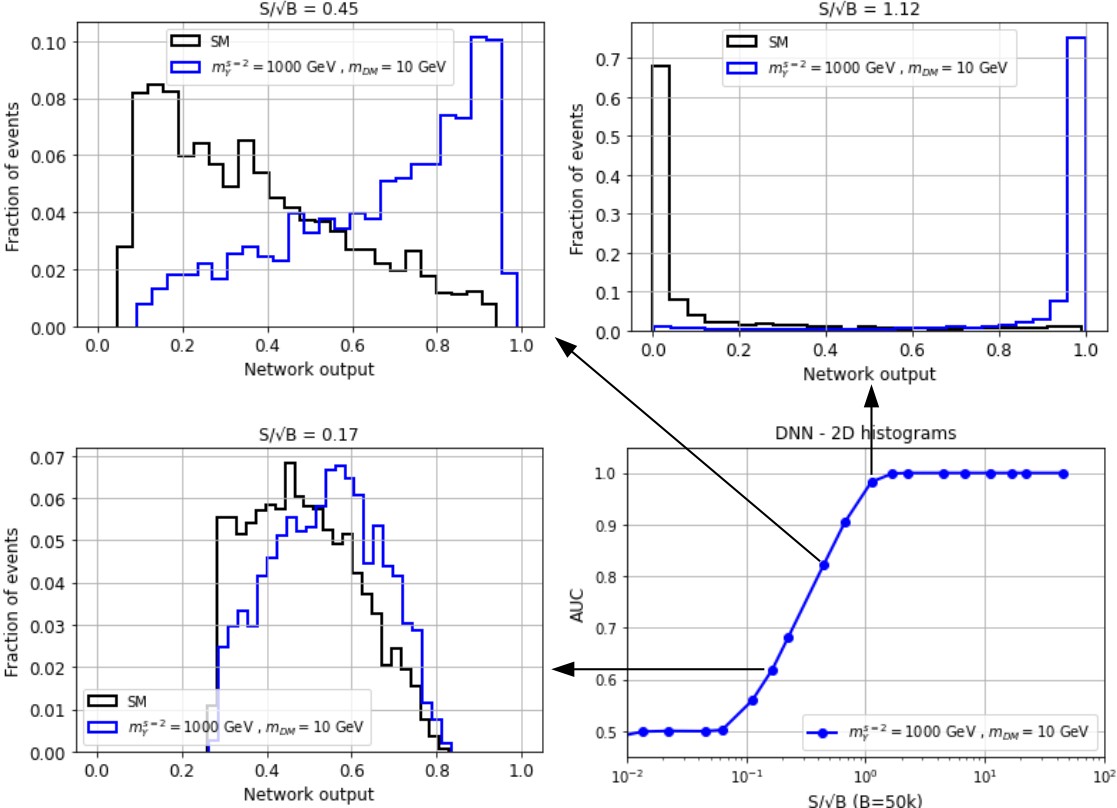

Figure 8: Network outputs for $S/\sqrt{B} = 1.12$ (top-right), 0.45 (top-left) and 0.17 (bottom-left), considering the benchmark model with $m_Y^{s=2} = 1000$ GeV and $m_{DM} = 10$ GeV, whose performance and AUC values can be seen on the bottom-right panel. The outputs black (blue) curves represent the results when test samples of SM-only (new physics plus SM) histograms are applied.

are applied, and the blue curves show the network outputs for test samples of histograms containing new physics plus SM events (recall that we expect '0' and '1' for the former and latter classes of histograms respectively, as stated in the first column of Table 1). As anticipated by the AUC results, the output distributions are more separated, and therefore easier to discriminate, for higher values of $S/\sqrt{B}$. To assign a label and classify each histogram, one has to define an output threshold considering the signal acceptance or background rejection efficiency to work with. Then, histograms with outputs below (above) this threshold are labeled as '0' ('1'), i.e. the network predicts that the histograms were constructed without (with) one or more new physics events.

When dealing with experimental data, the usage of DNNs with histograms is best suited as a fast initial analysis. The SM background for the working luminosity with our very simple cuts could be estimated, then the maximum number of DM events compatible with the data could be found. The experimental events could be arranged into a single histogram which would be tested by a DNN, corresponding to a particular benchmark model with the compatible $S/\sqrt{B}$. Naturally, other scenarios with smaller $S$, or different couplings and masses, could also be compatible and tested. As explained before, an output threshold has to be establish to determine if the experimental data is labeled as '0' or '1'. If the network returns a positive result, i.e. the DNN points towards the presence of new physics within the data set, a specific

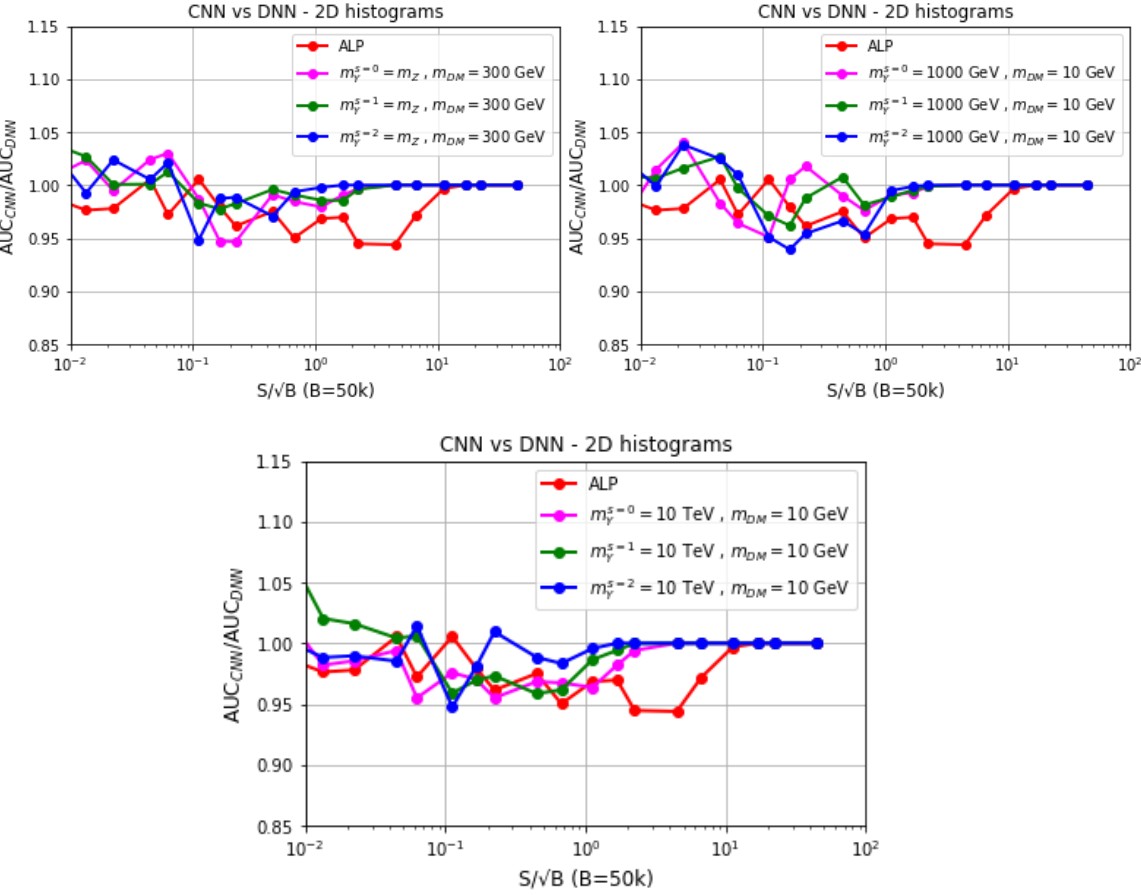

Figure 9: CNN AUCs divided by the DNN AUC for the benchmark models considered, using 2D histograms as input data. Each model is tested independently in a binary classifier to distinguish new physics plus SM vs SM-only histograms.

counting analysis should be applied to establish the significance of the candidate signal.

Finally, we would like to highlight that some points in Fig 7 are likely already excluded by current monojet searches, especially for high $S/\sqrt{B}$ values. Our main focus is to assess the network performance and behavior in a general way, hence determining the constraints for each particular model is out of the scope of this work.

### 3.2.2 CNNs with data as histograms

In this subsection we consider convolutional neural networks (CNN) as individual binary classifiers using the 2D histograms, constructed with monojet kinematic data as input images. The AUC obtained with CNNs is compared with the DNN results to contrast the performance of each algorithm. As before, data samples are divided with a 0.64:0.16:0.20 train-validation-test ratio. More details of the CNN structure can be found in Table 2.

In Fig. 9 we show the CNN AUCs divided by the DNN AUC obtained in the last section. Notice that the differences are within $\sim 5\%$ for all the models considered, meaning no improvement with respect to the DNN is found. Due to the characteristics of our data this is expected, since the method can not use one of the CNNs most important features: shift or space invariance, i.e. if we translate the inputs ('moving' the image) the CNN will still be able to detect the class to which the input belongs. This property is useful when the exact location

of the object is not required. However, our data set (the histograms) does not undergo translations by construction. As CNNs also demand significantly more computing time, we focus on DNN implementation for the rest of this work.

# 4 Flexibility of the method

In this section we will study how flexible and robust the method involving histograms is. First, we show that the DNN performance does not depend on the simulated number of background events. AUC curves turn out to be stable if we choose $S/\sqrt{B}$ as variable. Then, we analyze the DNN performance when incorrect assumptions are considered. We apply to an already trained DNN, a test data sample generated with a different model than the one it was trained. Two key features are tested: i. discrepancies in $S/\sqrt{B}$, equivalent to consider different coupling values, and ii. different benchmark models (dark sector masses and mediator spin), which allow us to see if new physics can be discriminated from the SM background even when the underlying theory is not the one we are assuming. Finally, we study the discrimination power among different DM models.

In Appendix D we discuss other neural network details with 2D histograms: no significant difference in performance with parton and detector-level simulated data, the impact of bin size, linear vs logarithmic representation to define those bins, and different $p_T^j$ selection cut values.

## 4.1 Performance stability with the simulated number of background events

In previous sections, for every example shown we considered 2D histograms constructed with $B = 50k$, a value that depends on the collider luminosity as the SM monojet cross section process is well defined by the cuts considered. However, we will see that the DNN performance is not modified significantly for different values of $B$, if the results are presented as a function of $S/\sqrt{B}$.

In Fig. 10 we show the AUCs results for 2D histograms constructed with $B = 20$, 100, 1k, 5k, 10k and 100k, as indicated on the figures, divided by the AUC considering $B = 50k$. Notice that differences are within $\sim 5\%$ for the benchmark models considered, unless the value of $B$ is not large enough to represent the underlying distribution, characteristic that is analyzed by the algorithm. In our examples, we have found that for $B \sim 100$ the differences begin to be $\gtrsim 10\%$, however these values are much smaller than the typical number of expected monojet events. Therefore, for sufficiently large values of $B$ ($\gtrsim 1k$) the performance is very similar to the corresponding cases with $B = 50k$ presented in Fig. 7.

This crucial property provides flexibility and robustness to the method, allowing us to evaluate the algorithm performance in a general way, regardless the size of the initial data, or even if a sub-set is selected. For example, if we would like to know if a DNN or CNN with 2D histograms could distinguish a particular new physics model from the SM background, first we must make sure that we have a good classifier, and for that we only need to

- identify the curve of the corresponding framework in Fig. 7,

- calculate the model cross section for the chosen couplings,

- calculate the SM-background cross section,

- calculate $S/\sqrt{B}$ for any luminosity, and check the corresponding AUC.

Throughout the rest of this work, we are not going to specify the value of $B$ in the figures. We note here that we use histograms with $B = 50k$ to obtain the results in all cases.



Figure 10: AUCs for histograms constructed with $B = 20, 100, 1k, 5k, 10k$ and $100k$, as indicated on the figures, divided by the AUC considering $B = 50k$, for several benchmark models. Results for the case with $B = 50k$ can be found in Fig. 7. Each model is tested independently in a binary classifier to distinguish new physics plus SM vs SM-only histograms.

## 4.2 Testing under incorrect training hypothesis: coupling values

To continue with the analysis of the DNN robustness in the search of DM, in this and the next subsection we study the DNN performance when incorrect assumptions are considered.

It is important to highlight that, as explained before, we divide our simulated events or histograms in two data sets: train and test samples (to simplify the explanation, we consider the validation sample as part of the training data set throughout the rest of this work). As

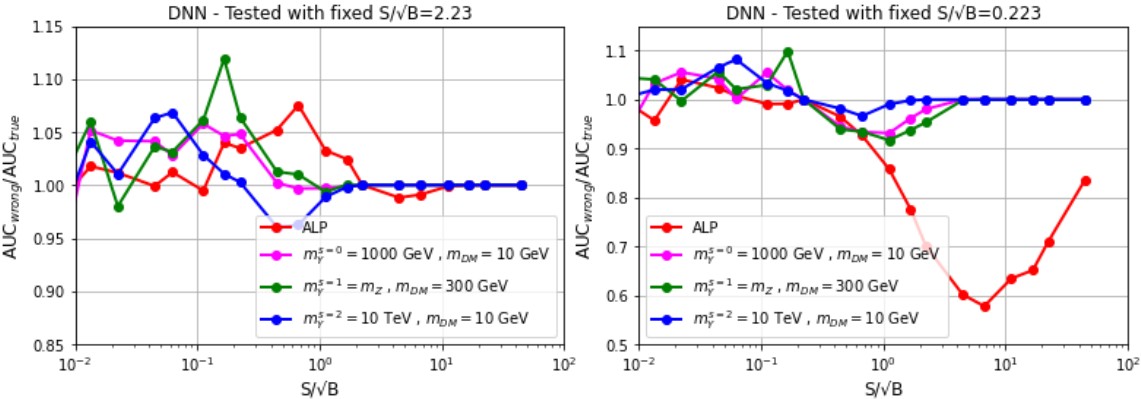

Figure 11: Ratio of the AUCs for histograms constructed with $S/\sqrt{B} = 2.23$ (left panel), and $S/\sqrt{B} = 0.223$ (right panel), applied on DNNs trained with $S/\sqrt{B}$ as pointed out on the horizontal axis. $\text{AUC}_{\text{wrong}}$ corresponds to the result of DNNs with non-matching test and train data sets, and $\text{AUC}_{\text{true}}$ with matching samples.

expected, a DNN is prepared to handle the same kind of data that it has been trained with. To find out its performance, we apply to an already trained DNN, a matching test sample, i.e. generated with the same underlying model as the train one, and calculate the AUC. For example, if we train a network with ALP and SM histograms with $S/\sqrt{B} = 2.23$, the DNN is prepared to test or classify new ALP and SM samples with $S/\sqrt{B} = 2.23$. DNN results with matching data samples are denoted $\text{AUC}_{\text{true}}$, and were shown in Fig. 7 for the benchmark models.

The next question arises, what is the performance when we try to classify a test data set with a DNN prepared to handle another kind of data, i.e. trained with a data set generated with a different underlying hypothesis than the test sample? DNN results with non-matching data samples are called $\text{AUC}_{\text{wrong}}$.

In this case, train and test data samples are generated with the same benchmark model, but with different $S/B$ ratios, i.e. a same model with different coupling values. Two examples are shown in Fig. 11, on the left panel the test data set consist of 2D histograms with fixed $S/\sqrt{B} = 2.23$, and on the right panel to $S/\sqrt{B} = 0.223$. To summarize, each point shows $\text{AUC}_{\text{wrong}}/\text{AUC}_{\text{true}}$ when we apply the test data set with a fixed $S/\sqrt{B}$ value, to a DNN trained with a sample whose $S/\sqrt{B}$ value is shown in the horizontal axis. Four benchmark models are shown as an example, but the same conclusions can be found for the other ones.

We can see that around the matching $S/\sqrt{B}$ the performance of the DNN evaluated with an incorrect signal-to-background ratio is close to the properly evaluated DNN. Although each DNN is trained for a particular model, the algorithm is flexible enough to handle some error or variations in $S/\sqrt{B}$, i.e. in the coupling values. Naturally, if train and test data samples that are constructed with very different ratios, the DNN predictions are not reliable.

We would like to remark that for $S/\sqrt{B} = 2.23$, and 0.223, each model has a $\text{AUC}_{\text{true}}$ different to 0.5 and 1, at least in one benchmark model (see Fig. 7). For the former $S/\sqrt{B}$ value, we get $\text{AUC}_{\text{true}} \sim 1$ for the benchmark models with a mediator, and $\sim 0.75$ for ALPs. In this high signal-to-background scenarios the test sample histograms applied to a DNN trained with a different $S/\sqrt{B}$ ratio, have enough signal events to keep the original performance in the entire range. As can be seen on the left panel of Fig. 11, the $\text{AUC}_{\text{wrong}}/\text{AUC}_{\text{true}}$ ratio is within $\sim 10\%$.

On the other hand, for the ALP case with $S/\sqrt{B} = 0.223$ we get $\text{AUC}_{\text{true}} \sim 0.5$ (see Fig. 7).

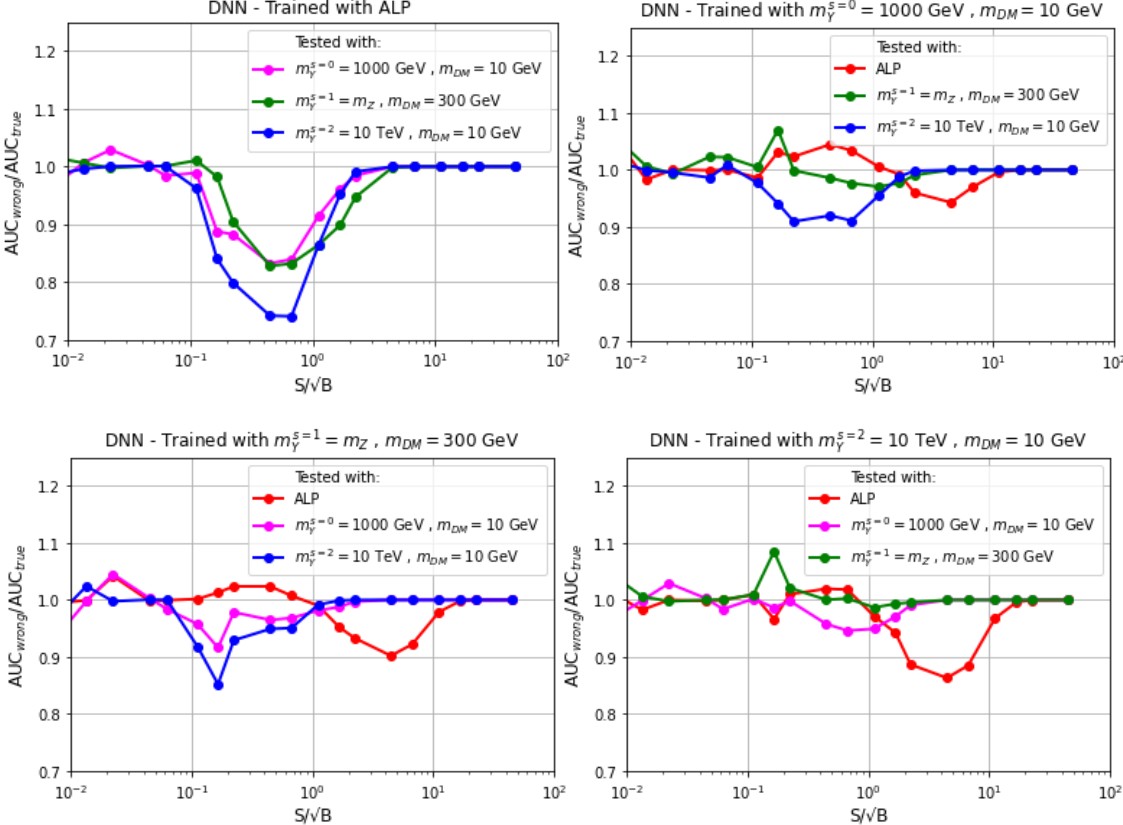

Figure 12: Ratio of the AUCs for non-matching train and test data samples, generated with different benchmark models. Test samples generated with several models, as indicated on each label, are applied to a DNN trained with a particular benchmark: ALP (top-left), spin-0 mediator (top-right), spin-1 mediator (bottom-left), and spin-2 mediator (bottom-right), as indicated on top of each panel.

Therefore, we are using a test sample without enough signal events to be easily handle by the DNN. When we test this kind of low signal histograms with a DNN that was trained with larger values of $S/\sqrt{B}$, significant differences can appear, as can be seen on the right panel of Fig. 11. However, if we test with a DNN trained with even larger values of $S/\sqrt{B}$ the performance improves. In this scenario, the DNN has enough training data to determine more accurately the underlying distributions, and starts to distinguish better the test data with low signal histograms.

## 4.3 Testing under incorrect training hypothesis: benchmark model

Next, we study the DNN performance when an incorrect benchmark model is used to generate the train data samples. We employ the same four benchmark models from the previous subsection, whose AUC$_{\text{true}}$ can be seen in Fig. 7. Results are shown in Fig. 12. Each DNN is trained with a particular benchmark model and tested with histograms constructed with a different benchmark, but with matching $S/\sqrt{B}$ values. On top of each panel the training model is indicated, and each curve is labeled with the test model. We can see that, for high $S/\sqrt{B}$ values we get good discrimination power and the performance is similar to the original one, since AUC$_{\text{wrong}}$/AUC$_{\text{true}}$ is within 10%. This is expected, because for histograms with large signal-to-background ratios there are plenty new physics events, and an easy discrimination

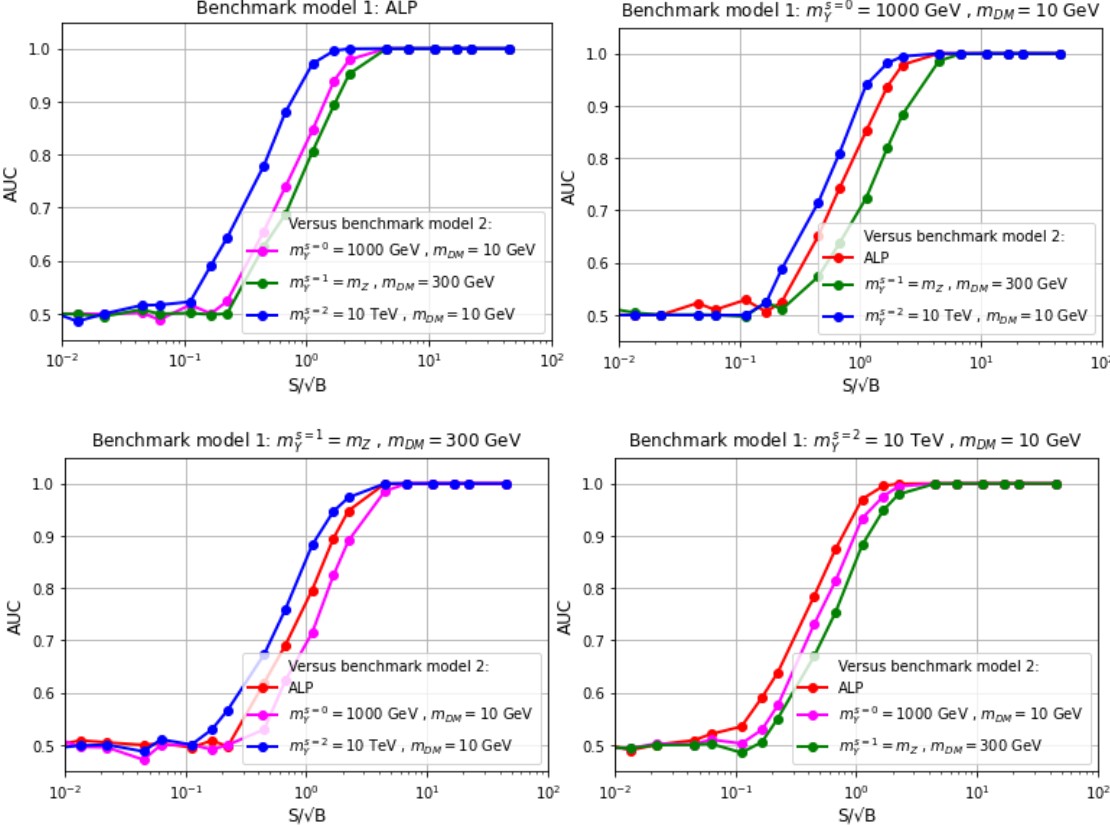

Figure 13: AUC for several binary DM classifiers. Each panel shows the efficiency when discriminating between a benchmark model 1 plus SM, versus a benchmark model 2 plus SM, as indicated on top and in the label of each figure correspondingly.

from the SM can be done regardless the true underlying model.

However, for intermediate $S/\sqrt{B}$ values, discrepancies can be significant, up to 25%, and the DNN predictions are not reliable. Notice that this is the same region for which the DNN results for matching test and train samples (see Fig. 7) lie between 0.5 and 1, i.e. when its intermediate signal-to-background ratios results in a harder, but still possible, discrimination. In that sense, to obtain a good performance it is important to test a DNN with the correct model, or at least a model whose kinematic distributions are similar to the trained one.

Finally, for low $S/\sqrt{B}$ values the performance of the incorrect models is similar to the correct one, but recall that in this region $\mathrm{AUC_{true}} \sim 0.5$.

## 4.4 Discerning between DM models

If a monojet signal analysis indicates the presence of non-SM processes, for example, with the method proposed in previous sections, we would like to identify the underlying new physics framework. Therefore, we explore the DNN performance to distinguish among different DM benchmark models. We consider DNNs with the same structure as before (see Table 2 for more details), and use the 2D histograms constructed with monojet kinematic data, unrolled as 1D arrays as inputs.

We employ binary classifiers, but instead of trying to discriminate between a benchmark model plus SM versus SM-only histograms, in this subsection we use the DNNs to distinguish between a benchmark model 1 plus SM, versus a benchmark model 2 plus SM histograms. We

analyze several new physics models whose performances are shown in Fig. 13. Each panel presents the AUC when discriminating between one benchmark model, as indicated on top of each figure, versus a different one, as can be seen on the labels. In all cases an individual DNN was trained for each pair of benchmarks, with matching train-test data samples.

On the top-left panel, we show the performance distinguishing ALP versus the other models, individually. We can see that the results are very similar to the ones shown in Fig. 7, where each model is tested against the SM background. In this case, the ALP model takes the role of the SM monojet, because the kinematic distributions of both frameworks are similar.

On the top-right and bottom-left panels, we show the performance for spin-0 mediator and spin-1 mediator respectively, versus the rest of the models, individually. Spin-0 and spin-1 mediator models have similar kinematic features, and therefore, present similar results when are contrasted against other models. Also, notice that the lowest efficiency is achieved when we try to discriminate between spin-0 and spin-1 mediators themselves. Nonetheless, the DNN can handle these similarities and we get AUC $\gtrsim 0.9$ for $S/\sqrt{B} \gtrsim 2.5$.

On the bottom-right panel, we show the AUC from DNNs trained with the spin-2 mediator vs the other models, individually. The spin-2 model has the hardest kinematic distribution of all the benchmark models probed, hence the highest efficiency is achieve when we try to disentangle between the spin-2 model and the model with the softer distribution, ALP.

To conclude this subsection, we would like to mention that good performances are found with the 2D-histogram approach and discrimination between different DM benchmarks is possible in a large range of $S/\sqrt{B}$. As we have seen when trying to discern between new physics and SM, these performances could not be achieved with the event-by-event approach.

# 5 Multimodel classifiers

In previous sections we discussed the identification of different new physics monojet signatures with respect to the SM background or another DM model, training each DNN with one or two DM benchmarks. Therefore, we end up with several DNNs per signal-to-background ratio (one per benchmark model), with good performance, but each one needs computer time for training and in principle only can test similar models to the one with which they have been trained.

Given the power and flexibility of DNNs so far, we would like to have a single DNN per $S/\sqrt{B}$ value, able to discriminate between SM background and non-SM processes, regardless the underlying model. Hence, we study the performance of a single DNN trained with several new physics models.

We study two multimodel ML methods, binary and multiclass classifiers. In the former, we train a DNN to distinguish between SM background and several new physics processes. In the latter, we take one more step, and train a DNN to identify the underlying DM model among those considered. For both methods, we analyze the efficiency when tested with trained and non-trained models.

## 5.1 Binary classifier

We consider DNNs with the same architecture as shown in Table 2. We use the 2D histograms constructed with monojet kinematic data, unrolled as 1D arrays as inputs, with SM background and seven benchmark models, as can be seen in the second column of Table 1. To evaluate the DNN performance under new unexpected data, three benchmark models are not used in the training set and therefore do not have a label assigned. Then, as training data we have two examples of each spin, and two examples of each regime (*on-shell, off-shell, off-shell PS*).

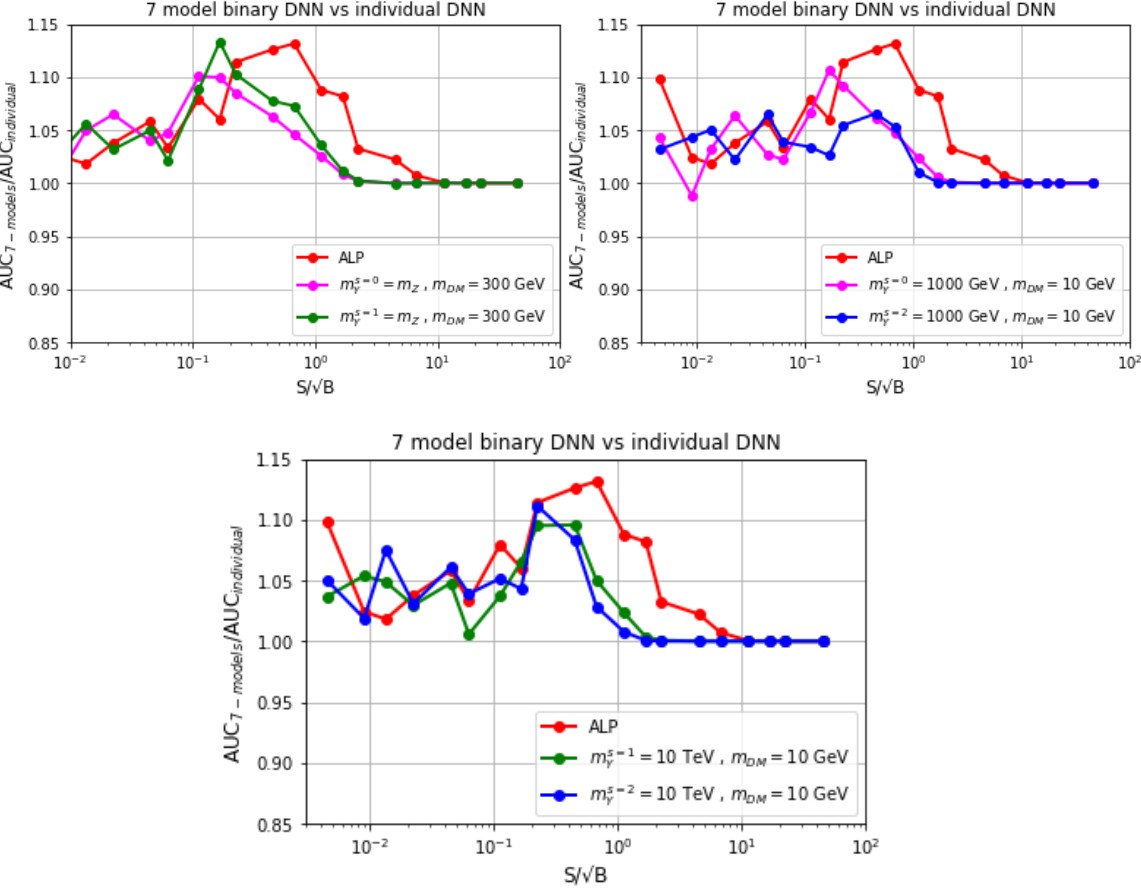

Figure 14: AUCs obtained by applying a test sample of a single model to a DNN trained with 7 different models, AUC$_{7\text{-models}}$, divided by the AUCs obtained by applying the previous test sample to a DNN trained with a matching data set, AUC$_{\text{individual}}$. Each model is tested independently in a binary classifier to distinguish new physics plus SM vs SM-only histograms.

Each 2D histogram constructed with only SM-background events is labeled with a '0', while every other histogram containing non-SM events is labeled with a '1'. Therefore, we want to discriminate between SM-only and any kind of new physics plus SM histograms. Notice that, as before, the samples are constructed with only one new physics model at a time, i.e. we do not mix different DM models in a single histogram. The DNN is trained with 10k histograms per model and per $S/\sqrt{B}$ value (and 10k histograms with only SM monojet processes).

To determine the efficiency of the DNN trained with multiple models, we apply a test data sample generated with only one benchmark model. We call this performance AUC$_{7\text{-models}}$, and the results are shown in Fig. 14. The performance of each model tested is divided by AUC$_{\text{individual}}$, i.e. the performance of the DNN trained and tested with matching data samples, i.e. both generated with a single benchmark model (the individual DNN performances shown in Fig. 7). We would like to remark again that the multimodel DNN is trained with 7 benchmark models, and the individual DNN with only one.

We find that some improvement, up to 15%, is achieved when we test each benchmark model with the 7-model trained DNN. As we are only training the network to distinguish between all new physics and SM-only background, the variety of models during the training stage helps the overall performance. However, the performance improvement ($\gtrsim 5\%$) occurs

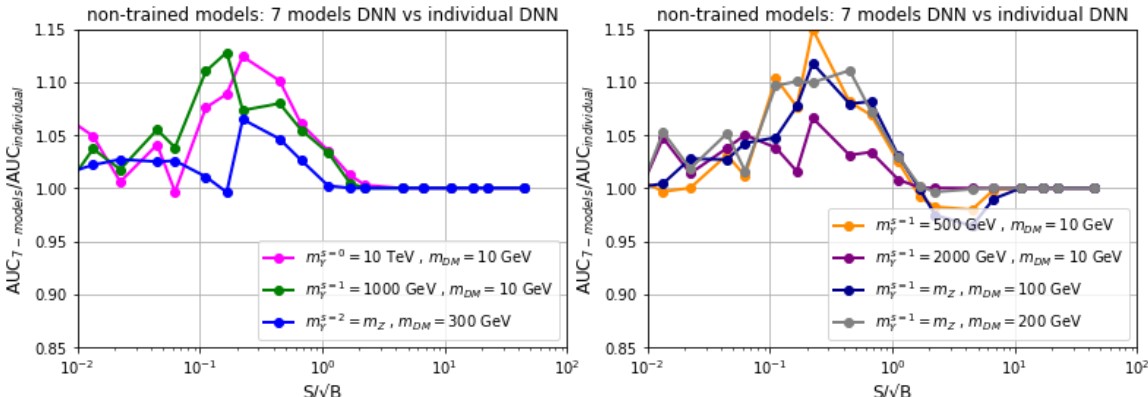

Figure 15: Same as in Fig. 14, but in this case test samples are generated with different models from those the network was trained with.

for $\text{AUC}_{\text{individual}} \lesssim 0.75 - 0.8$, i.e. when the DNN efficiency starts to be poor. It is important to notice that in this figure we test data samples generated with the same 7 benchmark models with which the DNN has been trained.

We can also analyze the 7-model trained DNN performance when we apply a test data sample generated with a model that was not used to generate the training sample. This is similar to the analysis done in Section 4.3, but with a multimodel binary classifier performing better than the single model binary classifiers. On the left panel of Fig. 15 we show the results for the three benchmark models that have not been included into the 7-model train data set, and on the right panel two extra on-shell models and two extra off-shell model are shown. The AUCs of each model has been divided by the performance of individual DNN classifiers with matching train and test data samples. As before, the difference among the AUCs is $\lesssim 15\%$, outperforming the previous method analyzed, and the performance improvement ($\gtrsim 5\%$) occurs when the individual DNN efficiency starts to be poor ($\text{AUC}_{\text{individual}} \lesssim 0.75$).

In summary, training a single DNN with data samples from a set of models provides a very useful way to test the DM hypothesis. Not only the 7-model DNNs reproduces the performance of individual DNN classifiers with matching train and test data samples for the 7 benchmark models selected, but also of several new models for which it was not prepared.

## 5.2 Multiclass classifier

To study the performance of a multiclass classifier, we use the same 7 benchmark models as in Section 5.1. However, each 2D histogram containing new physics events is assigned with a label from '1' to '7', to distinguish between the different benchmarks considered. Histogram constructed with SM background only are labeled '0', and take the role of an extra class. The labels are specified in the third column of Table 1. As before, three benchmark models are not used in the training set and do not have a label assigned to evaluate the DNN performance under new unexpected data.

The DNNs have the same structure as in previous sections (see Table 2 for more details), but use *softmax* as activation function in the output layer. In this way, the network output is an array containing eight elements (1 per model plus an extra one corresponding to the SM-only case) stating the probability that each histogram belongs to each class or model.

To start the performance analysis, we test the DNNs eight times, using the data set generated with the eight training models, individually. To determine the ROC curves and its corresponding AUCs, first we must convert the output array in a binary result: we only con-

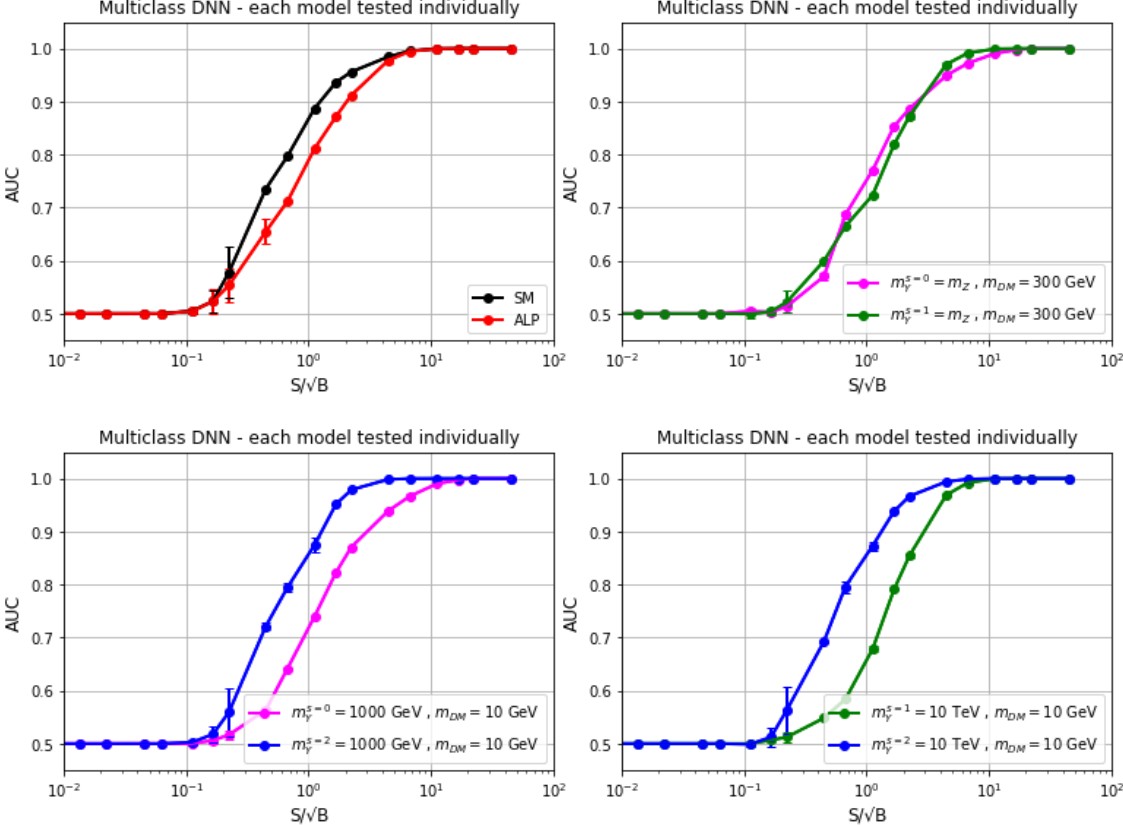

Figure 16: AUCs obtained by applying a test sample of a single model to the multiclass DNN trained with 7 different models and SM-only samples. The multiclass output array is converted into a binary result: we consider the element corresponding to the model we are testing as positive class, while the other elements are taken as negative classes. Error bars are included for all points.

sider as positive class, or signal, the element corresponding to the model we are testing, while the other elements are taken as negative classes. In Fig. 16 we show the results, with the standard deviation for every point, for $B = 50$k. As in Section 4.1, we have checked that the performance is not modified significantly for other values of $B$ when presented as a function of $S/\sqrt{B}$. We can see that the AUC value monotonically increases with the signal-to-background ratio, and that the algorithm can predict the correct benchmark model even for low $S/\sqrt{B}$ values. We would like to remark that in this section, the discrimination power is between a particular model and the rest of the 7 selected models (recall that in Fig. 7 the discrimination is between a particular model and SM-only hypothesis, so both figures can not be compared directly).

Furthermore, we can define another way to represent the DNN results to analyze which of the training models is predicted. One count is assigned to the output array element if its probability fulfills the following two conditions: it is the element with the highest value, and its value is above a threshold equal to 0.25, defined as two times the probability that would be obtained for a completely random classifier, i.e. $2/\#_{\text{elements}}$ (recall that the number of elements is 8, the 7 selected models plus the SM one). Otherwise, the element is assigned zero counts. Then, a histogram of the frequency of occurrence can be constructed. This representation is useful to check if, for a particular test data sample, two or more models are being predicted

Figure 17: Models predicted by the multiclass DNN applying test samples generated with the eight selected training models (see Table 1). Results for histogram samples with three signal-to-background values are shown, $S/\sqrt{B} = 6.7, 1.7$ and $0.45$

.

with similar probabilities, or which model is being chosen incorrectly if the DNN is performing poorly. However, ROC curves can not be constructed.

In Fig. 17 we show the results for three signal-to-background values $S/\sqrt{B} = 6.7, 1.7$ and 0.45, in blue, green, and red, respectively. We can see that for high $S/\sqrt{B}$ almost all the histograms are identified correctly. For intermediate $S/\sqrt{B}$ the most predicted model is still the correct one, but in some cases a significant fraction of events are labeled as a different model. This happens when two or more kinematic distributions are similar, for example SM and ALPs, then the DNN is not able to discriminate among those models efficiently with the mentioned $S/\sqrt{B}$ ratio. Finally, for low $S/\sqrt{B}$ only a small fraction of events fulfills the conditions mentioned before regarding the probability predicted, meaning that the DNN is not able to identify any model, and in the end an almost equal probability $\sim 1/8$ is predicted for each class.

Using the previous representation, test samples generated with non-trained models can also be analyzed. In the first and second row of Fig. 18 we show the results for the three benchmark models not used in the DNN training. Notice that, for this particular choice of train and test samples, the DNN predicts models with the correct mediator spin, but with an incorrect regime, i.e. wrong mediator status, *on-shell*, *off-shell*, or *off-shell PS*. On the other hand, this does not happen in the third and fourth rows of Fig. 18. We show there the results for test data samples generated with spin-1 mediator models: the first two are *on-shell* cases with different mediator masses, and the last two cases present an *off-shell* mediator with different DM masses. For example, the multiclass DNN predicts ALPs (label 1) for $m_Y^{s=1} = m_Z$ and $m_{DM} = 100$ GeV test samples (fourth row left panel). However, the DNN provides a better solution when we test $m_Y^{s=1} = m_Z$ and $m_{DM} = 200$ GeV data (fourth row right panel), predicting a similar model but with $m_{DM} = 300$ GeV (label 4).

After all, the DNN is working with 2D histograms that represent the underlying kinematic distributions of each model. If we compare the test and predicted models in Fig 18, with the $p_T^j$ distributions of the corresponding models shown in Section 2.3, we can see that the DNN is assigning the label that better matches these features.

In summary, multimodel multiclass classifiers can be very useful when we need to discriminate a possible signal among a subset of specific new physics models. We have shown that if we test different models from those the network was trained with, the results can be misleading if we want to extract information about the parameters of the models, for example the mediator spin, or the DM mass. Nonetheless, in those cases the DNN provides crucial information, identifying the compatible kinetic distribution of the underlying model.

# 6 Discussion

In this work we make use of machine learning techniques in the search for new physics. In particular, we focus on the search for dark matter signatures at the LHC using deep neural networks. We employ supervised algorithms which allow us to obtain large performances but implies that specific models have to be considered and big labeled data samples have to be constructed.

To be as general as possible, we study the monojet plus missing transverse energy channel of four simplified dark matter frameworks, ALP and spin-0, spin-1, and spin-2 mediator models, described in Section 2.1. One important characteristic of our approach is the use of kinematic features as DNN input data. We have seen that the kinematic signatures do not depend on framework coupling values and type of DM candidate. This allows us to overcome one usual drawback of supervised techniques, the need of a specific data set per model, by describing a family of models with a single data sample.

Nonetheless, in the case of frameworks with a mediator we still have two free parameters, the masses of the dark sector particles. This results in different kinematic signatures and therefore each one has to be probed separately. We found out that if we consider the status

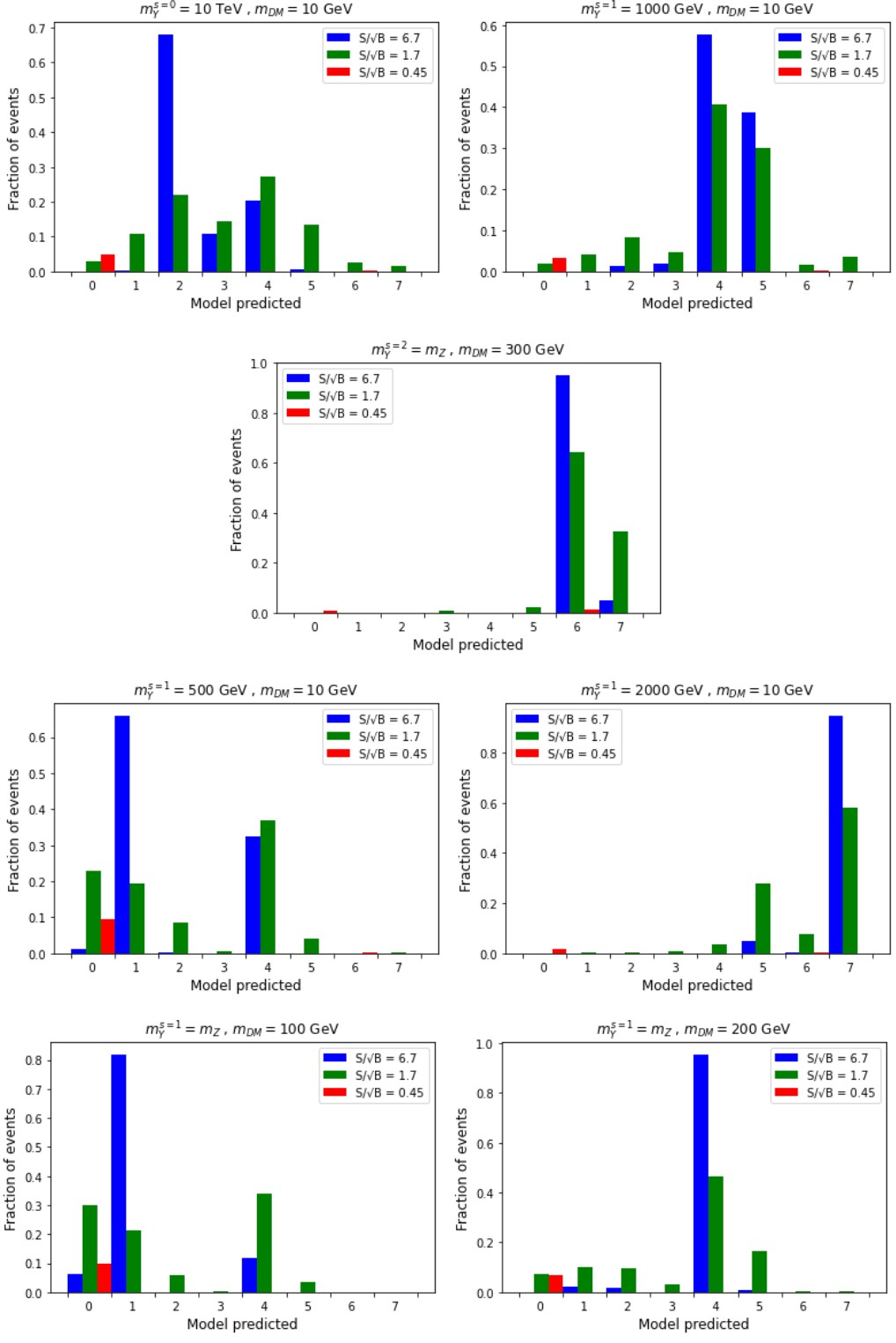

Figure 18: Same as in Fig. 17, but test samples are generated with different models from those the network was trained.

of the dark mediator (off-shell, on-shell, and off-shell by phase space), only one parameter describes the overall shape of the kinematic distributions, either the mediator or the dark matter mass. Taking this into consideration, we can define a reduced number of benchmark models describing the most general characteristics of each framework to be tested with LHC data.

Regarding the data analysis with DNN, we have seen that the data representation has a strong impact on the performance of the algorithms. First we train and test our DNNs with event-by-event simulated data and found mediocre performances discerning new physics signatures from SM background (see Fig. 5). Later, we organize the kinematic features in 2D histograms considering the jet transverse momentum and pseudorapidity, and found a large boost in the discrimination efficiency, up to 1 for high enough signal-to-background ratios (see Fig. 7). We could also use the 2D histograms as images for a convolutional network. However, a key CNN characteristic is not used, the space invariance, as our data set does not undergo translations. Hence, no improvement with respect to DNN is found.

One potential drawback of the histogram representation may lie in the construction of the histograms themselves, given that each one is generated with $S$ new physics events and $B$ SM-background events. Therefore, several data sets per model have to be generated to take into account that different coupling values can generate different amount of events for a given collider luminosity. Although at first glance it seems that we reintroduce the coupling values as parameters, the only important quantity is the number of new physics events, $S$, since different coupling values, as well as the dark sector masses under certain conditions, do not change the underlying kinematic distributions (see Section. 2.3). Then, each histogram with $S/B$ ratio describes a family of models.

A crucial feature of the method is that the DNN performance turns out to be independent of the simulated number of background events, for $B \gtrsim 1$k (value much smaller than the expected number of monojet events). The AUC, a conventional metric of the DNN performance, is stable when we choose $S/\sqrt{B}$ as variable (see Fig. 10). Therefore, to find out if a DNN with 2D histograms would be a good classifier to distinguish new physics from the SM background, the reader has to follow four simple steps:

- identify its model with one of the benchmark models used, taking into account that the kinematic distributions have to be similar,

- calculate the SM-background cross section, $\sigma_{\text{SM}}$,

- calculate the model cross section with the particular coupling values that the reader wants to test, $\sigma_{\text{NP}}$,

- calculate $S/\sqrt{B}$ for the luminosity available, and check the corresponding AUC in Fig. 7.

Moreover, we established that the method is quite robust. As said before, one drawback of the supervised algorithms is that, in principle, they are only prepared to handle the same kind of data with which they were trained. However, when we use non-matching test and train samples, we found out that the performance is not modified significantly in two scenarios:

- if train and test histograms are generated with the same model, but different coupling values are considered, i.e. variations in $S/\sqrt{B}$, as long as those signal-to-background ratios are not very different (see Fig. 11),

- if train and test histograms are generated with different benchmark models, as long as their kinematic distributions are similar (see Fig. 12). In this category we include small variations in the dark sector masses.

This kind of DNN can also be used to disentangle among different DM models, if a new physic signal is found. The 2D-histogram representation presents good performances, and can even classify benchmark models with similar kinematic distributions using DNNs trained to discriminate between a specific pair of models as discussed in Section 4.4.

At this point we would like to highlight that we can describe a large number of DM scenarios with a handful of benchmark models. Focusing on the monojet kinematic features allow us to describe a family of models with a single data sample. The 2D-histogram representation is key to obtain good performances, however we still need to train one DNN per benchmark model and per $S/\sqrt{B}$ value. Although the DNN is flexible enough to handle some variations in coupling values and dark sector masses, we still need to apply our data to a DNN trained with similar characteristics. However, this can be very challenging if we do not know anything about the true underlying framework present in nature.

To overcome this issue we use multimodel classifiers, supervised algorithms but instead of training one benchmark model per DNN, several ones are used. In Fig. 14 we employ a multimodel binary classifier prepared to discriminate between SM-only histograms and data with SM background plus several models of new physics events. It is remarkable that we get a small improvement of performance with respect to DNN trained with one benchmark model at a time, even for models not included in the training list.

In Section 5.2 we described a multiclass DNN prepared to identify between SM-only and several benchmark models, pointing out the most likely underlying model. Even though this turns out to be a more challenging task, a good performance is achieved. If trained benchmark models are tested, we get correct model identification for high and intermediate $S/\sqrt{B}$ ratios (see Figs. 16 and 17). However, for models not included in the training benchmark list, incorrect model properties (e.g. dark sector masses or mediator spin) are predicted as a consequence of not being in the database (see Fig. 18). Despite that, in those cases the DNN can still discriminate new physics from the SM-only hypothesis, and provides crucial information about the true underlying model. The multiclass DNN result points towards a compatible kinetic distribution, a key tool to guide further analysis.

Finally, it is important to emphasize that the information provided by each representation discussed is different. The event-by-event method tries to determine whether each event comes from a SM or new physics process. On the other hand, the algorithms that classify histograms try to determine if there is new physics in a set of events (recall that each histogram is constructed with $S$ new physics events and $B$ background events). However, the latter method does not classify each individual event.

Given that we have used very simple cuts and the high performances that can be achieved, the usage of DNNs with histograms is best suited as an initial analysis. The DNNs performance turns out to be independent of the simulated number of background events, thus we can anticipate if a set of events contains new physics with a relatively fast and general approach. If the network shows a positive result, specific counting analysis guided by the information extracted with the histogram method should be applied to establish the significance of the candidate signal.

## 6.1 Event-by-event combination and the histogram method

Throughout this work we have classified sets of events by constructing histograms and training dedicated networks. As pointed out in Refs. [47,48], from a statistical point of view, the optimal histogram classifier could be built from an optimal event-by-event classifier. We have checked that combining the event-by-event result of each benchmark model studied in Section 3.1, we can obtain the same AUC curves shown in Fig. 7 using the histogram method. See Ref. [48] for more details about the procedure. In Fig. 19 we present one example considering the *off-shell* spin 1 benchmark point, and we can see that both methods yield the same

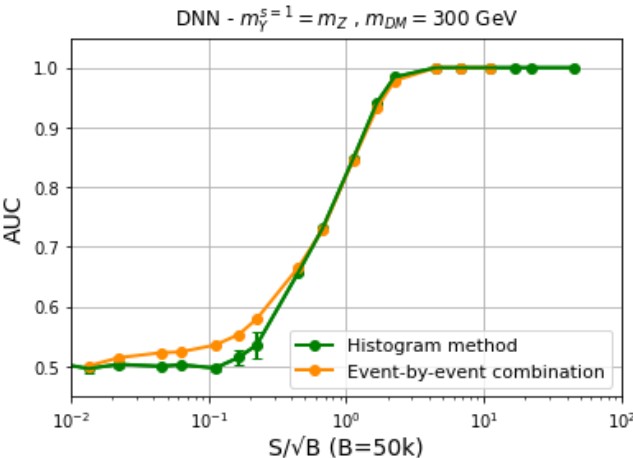

Figure 19: AUC results for the *off-shell* spin 1 benchmark point using the histogram method (same as in Fig. 7), and combining the outputs of the corresponding event-by-event DNN found in Section 3.1 (see Ref. [48] for more details).

results, except for low AUC values ($\lesssim 0.65$) where the performance of both methods is poor and classification is not possible.

This verifies that we have obtained the optimal histogram classifiers with the DNN architecture, parameters, and data samples that we have described. Although combining the event-by-event output is a priori a simpler procedure, we have found some issues to consider. First, small variations in the event-by-event training can produce significant variations in the deduced per-histogram performance, even decreasing the AUC by a few per cent. This discrepancy can be found for event-by-event networks with almost identical outputs, ROC curves and no overfitting signals that we can appreciate. Second, even though manageable this case, special care has to be considered when combining event-by-event outputs, specially for large number of events per histogram, as numerical problems can occur and produce misleading poor performances. In that sense, although the histogram method is computationally more expensive, in our particular case it seems to be more stable than the procedure described in Ref. [48], and therefore the optimal classifier can be obtained in a more straightforward way.

As expected, the aforementioned issues are even more relevant for increasingly complex scenarios, such as multimodel classifiers. As far as we know, combining the result of event-by-event classifiers is not trivial for those cases, but it is relatively simple with the histogram method shown in Section 5.

To conclude, we want to mention that an interesting alternative to tackle the problems found and to benefit from the simpler per-event training would be to train an event-by-event classifier but to monitor its performance by designing a specific per-histogram validation loss that considers sets of events. Then, the outputs of the resulting algorithm could be combined to obtain an optimal histogram classifier. We leave that analysis for future work.

## 7   Conclusions

In this work we have analyzed the performance of DNN to disentangle dark matter signatures from SM background at the LHC. We focused on the usual monojet plus missing transverse energy channel and explored different data representations. First we studied several simplified

DM models and found key parameters that allowed us to represent a family of models with a few data sets.

We saw that using the monojet kinematic features organized as 2D histograms improves significantly the DNN efficiency, but introduces extra parameters, namely the ratio between signal and background events, $S$ and $B$, respectively. Remarkably, the DNN results turns out to be independent of the simulated number of background events if they are represented as a function of $S/\sqrt{B}$, for reasonably large $B$ ($\gtrsim$ 1k). Furthermore, the method is flexible enough to handle small variations in coupling values and in other parameters, like the dark sector masses. All these properties make the DNN with histograms approach a good initial analysis to check in a fast and general way if new physics lie within a data set.

Finally, we explored two multiclass classifiers to ease the blind DM identification. The first one is a binary classifier prepared to distinguish SM-only histograms from samples with SM plus new physic events from several models. With the second method, a multiclass classifier, we took an extra step and is prepared to inform which one of the training benchmark models is more likely to represent the test sample, or if it only contains SM-background events. Both methods show good performances with high and intermediate signal-to-background ratios even when we test models for which the DNN has not been trained. Although the specific underlying model can not be reliably identified, crucial information about the kinematic distributions and therefore hints of the true model can be extracted.

# Acknowledgments

The authors would like to thank Ignacio Arganda-Carreras and Gonzalo Uribarri for helpful advises and insights on neural networks methods. We also thank Francisco Alonso, Fernando Monticelli, and Hernán Wahlberg for useful discussions about collider analysis and the method implementation. Last but not least, we would also like to express our gratitude to Benjamin Nachman and Jesse Thaler for very enriching and fruitful discussions and exchanges about event-by-event combination, the histogram approach and statistical interpretation. The work of EA is partially supported by the "Atracción de Talento" program (Modalidad 1) of the Comunidad de Madrid (Spain) under the grant number 2019-T1/TIC-14019 and by the Spanish Research Agency (Agencia Estatal de Investigación) through the grant IFT Centro de Excelencia Severo Ochoa SEV-2016-0597. This work has been also partially supported by CONICET and ANPCyT under projects PICT 2016-0164, PICT 2017-0802, PICT 2017-2751, PICT 2017-2765, and PICT 2018-03682.

# A  DM production channels

In Fig. 20, Fig. 21, Fig. 22, and Fig. 23 we show the monojet processes considered in this work for the ALP, spin-0, spin-1, and spin-2 mediator, respectively. For frameworks with a mediator, we find that the kinematic features are independent on the nature of the DM particle (real scalar, complex scalar, or Dirac fermion), therefore, the latter has not been specified on the figures.

For the spin-0 mediator framework the quark loop is dominated by heavy top-quark contributions due to the presence of SM Yukawa couplings.

For the spin-1 mediator framework, the diagrams are the same as in the usual SUSY Bino-like neutralino DM, $\chi_1^0$, replacing $DM \to \chi_1^0$ and $Y_1 \to Z$.

Finally, notice that spin-1 mediator channels shown in Fig. 22, and the first row of the spin-2 mediator channels in Fig. 23, have the same topology as the dominant SM-background

process $pp \rightarrow Zj(Z \rightarrow \nu\bar{\nu})$, replacing $DM \rightarrow \nu$ and $Y_{1,2} \rightarrow Z$. This particular diagrams are not present in the ALP and spin-0 mediator cases.

## B  SM-background processes

The primary SM-background contribution to monojet event signatures with missing transverse momentum in pp collisions at the LHC is

$$pp \rightarrow Zj(Z \rightarrow \nu\bar{\nu}). \tag{14}$$

The Feynman diagrams corresponding to the dominant process can be seen from Fig. 22, replacing $DM \rightarrow \nu$ and $Y_1 \rightarrow Z$.

There are also significant monojet contributions from [49]

$$pp \rightarrow Wj(W \rightarrow l\nu), \tag{15}$$

with non-identified leptons in the final state. Among these channels, the most significant contribution comes from tau leptons in the final state, $pp \rightarrow Wj(W \rightarrow \tau\nu)$.

Other small contributions are also expected, such as $pp \rightarrow Zj(Z \rightarrow ll)$, with $(l = e, \mu, \tau)$, multijet, $pp \rightarrow t\bar{t}$, single-top, and diboson $(WW, WZ, ZZ)$ processes. Finally, among the channels with negligible contribution we can find top-quark production associated with additional vector bosons $pp \rightarrow t\bar{t}W, t\bar{t}Z, tZq/b$.

To analyze how different channels contribute to the monojet signal, and their impact on our results using only the main SM background, we define the process relative contribution, $R$, as

$$R = \frac{N_i}{N_{pp \rightarrow Zj(Z \rightarrow \nu\bar{\nu})}}, \tag{16}$$

where $N_i$ is the number of events coming from a particular process and $N_{pp \rightarrow Zj(Z \rightarrow \nu\bar{\nu})}$ the number of events produced by the main SM channel. Both quantities are defined as

$$N_i = L\,\sigma_i\,\epsilon_i, \tag{17}$$

with $L$ and $\sigma_i$ the usual collider luminosity and process cross section at parton level, respectively, and $\epsilon_i$ the fraction of events that are identified as jets with missing transverse momentum at detector level, with respect to the total number of events simulated at parton level. Notice that in Eq. 16 the luminosity is canceled out.

In Table 3 we show the relative contributions of several processes (the multijet process has not been included as it has low $R$ [49]). We use the same tools described in the main part of this work, to generate events, to perform shower and hadronization, and to simulate the detector response. Also, we apply the same cuts to simulate parton-level data (for example, for $pp \rightarrow Wj(W \rightarrow l\nu)$ we use $p_T^l > 130$ GeV). For $pp \rightarrow t\bar{t}$, the fraction of events identified as jets and missing transverse energy is shown for three cases: no condition imposed on the final jets (first row), one $b$-jet or less (second row), and $b_{tag} = 0$ (third row). The results obtained are roughly compatible with the ones found by ATLAS Collaboration in Ref. [49], although different cuts and beam energies are considered.

As stated before, the second most significant contribution comes from tau leptons in the final state, $pp \rightarrow Wj(W \rightarrow \tau\nu)$, with $R \simeq 0.12$. The other channels are at the percent level, or less, due to the small value of $\epsilon$ coming from the miss-identification of final state particles as missing energy to obtain the required signal.

Finally, notice that $pp \rightarrow Wj(W \rightarrow l\nu)$ and $pp \rightarrow Zj(Z \rightarrow ll)$ processes are described by the same Feynman diagrams as the dominant channel, $pp \rightarrow Zj(Z \rightarrow \nu\bar{\nu})$, replacing $Z$ by

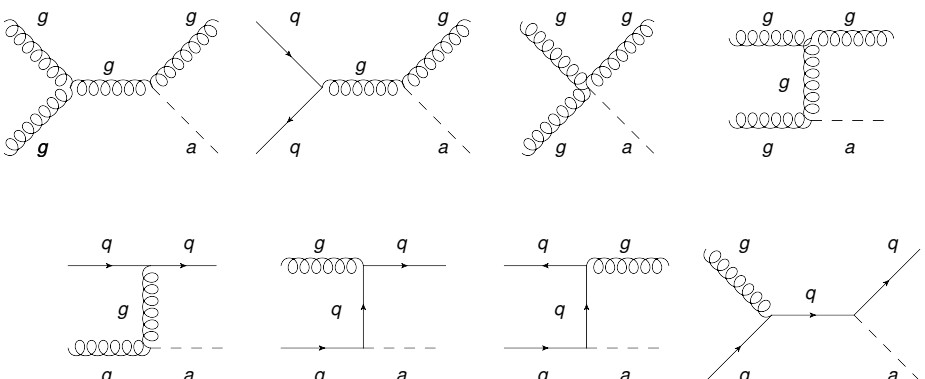

Figure 20: Feynman diagrams for DM production with a monojet in the ALP framework.

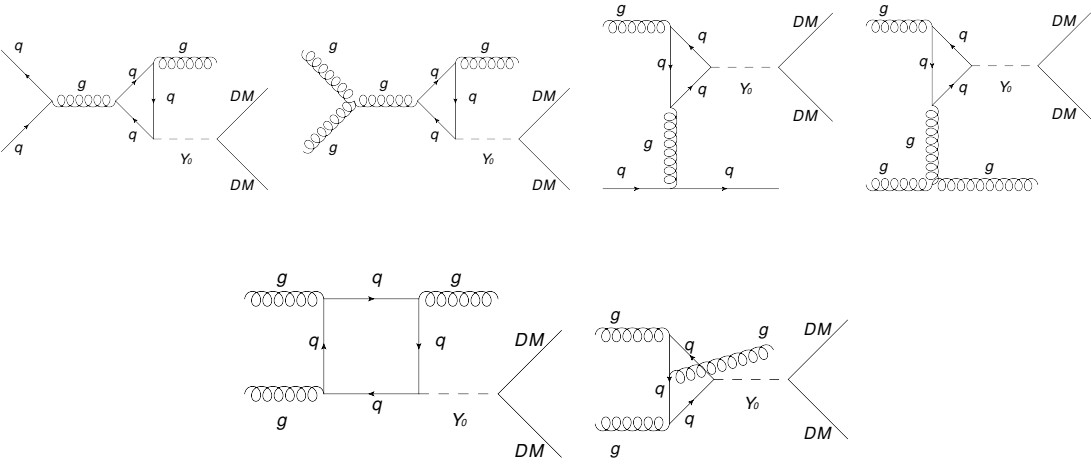

Figure 21: Feynman diagrams for DM production with a monojet in the spin-0 mediator framework. As mediator-SM quarks interactions are proportional to SM Yukawa couplings, loops are dominated by top-quark contributions.

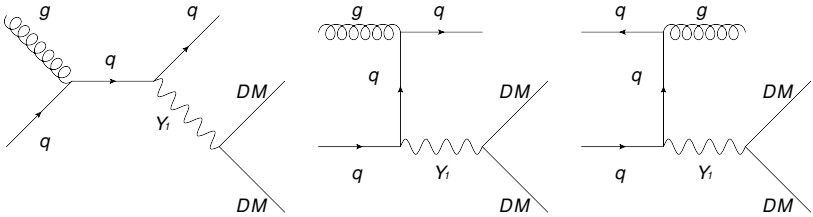

Figure 22: Feynman diagrams for DM production with a monojet in the spin-1 mediator framework.

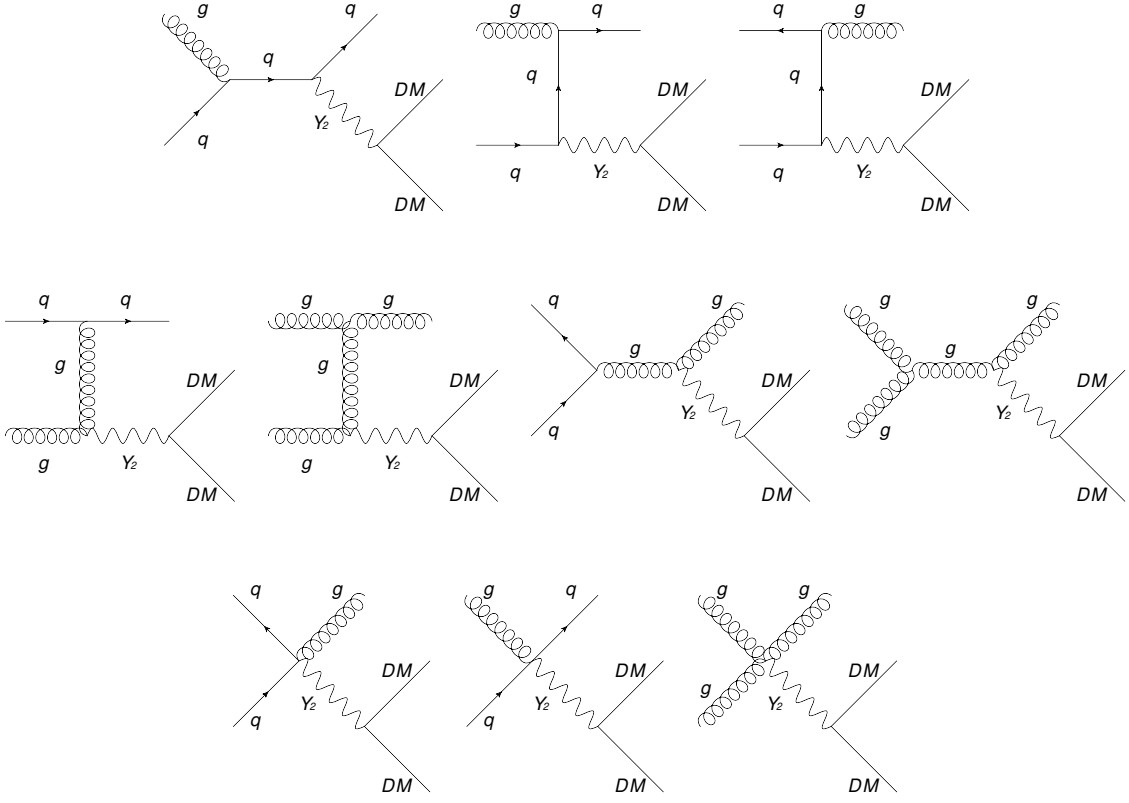

Figure 23: Feynman diagrams for DM production with a monojet in the spin-2 mediator framework.

$W$, and $\nu$ by $l$. Thus, these processes have the same kinematic distributions as the dominant channel when the leptons are misidentified. This is important because in this work we use jet kinematic features and distributions as input data for the ML algorithms. Even though to simulate the SM background, we use the dominant channel, our overall conclusions are not affected. The only effect that one has to keep in mind is that for a given luminosity, the total SM background is increased by the contribution of the secondary channels.

## C Spin-0 and spin-2 kinematic distributions

In Fig. 24 and 25 we show several $p_T^j$ distributions at parton level, for spin-0 and spin-2 mediator frameworks, respectively. The hardest and softest $p_T^j$ distributions that can be found are presented. The relevant parameter in each panel is $m_{DM}$ for the *off-shell* case (top-left panel), $m_Y$ for the *on-shell* regime (top-right panel), and $m_{DM}$ for the *off-shell PS* case (bottom panel).

Notice that the region covered by spin-2 mediator framework and shown in Fig. 25 is smaller than spin-0 and spin-1 cases. Also, for spin-2 there is no $(m_Y, m_{DM})$ value whose distribution is similar to the SM background.

Table 3: Process relative contribution ($R$) of several SM monojet plus missing transverse energy backgrounds, with respect to the dominant SM channel, $Zj(Z \to \nu\bar{\nu})$.

| Process | $\sigma$ (pb) | $\epsilon$ | $R$ |
|---|---|---|---|
| $Zj(Z \to \nu\bar{\nu})$ | 53.6 | 0.78 | 1 |
| $Wj(W \to \tau\nu)$ | 24.8 | 0.20 | 0.12 |
| $Wj(W \to l\nu), l = e, \mu$ | 49.7 | 0.049 | 0.058 |
| $Zj(Z \to ll), l = e, \mu, \tau$ | 19.1 | 0.013 | 0.006 |
| $t\bar{t}$ | 217 | 0.021 | 0.11 |
| | | 0.014 ($b_{tag} \leq 1$) | 0.073 |
| | | 0.004 ($b_{tag} = 0$) | 0.021 |
| diboson ($WW, WZ, ZZ$) | 5.18 | 0.12 | 0.014 |

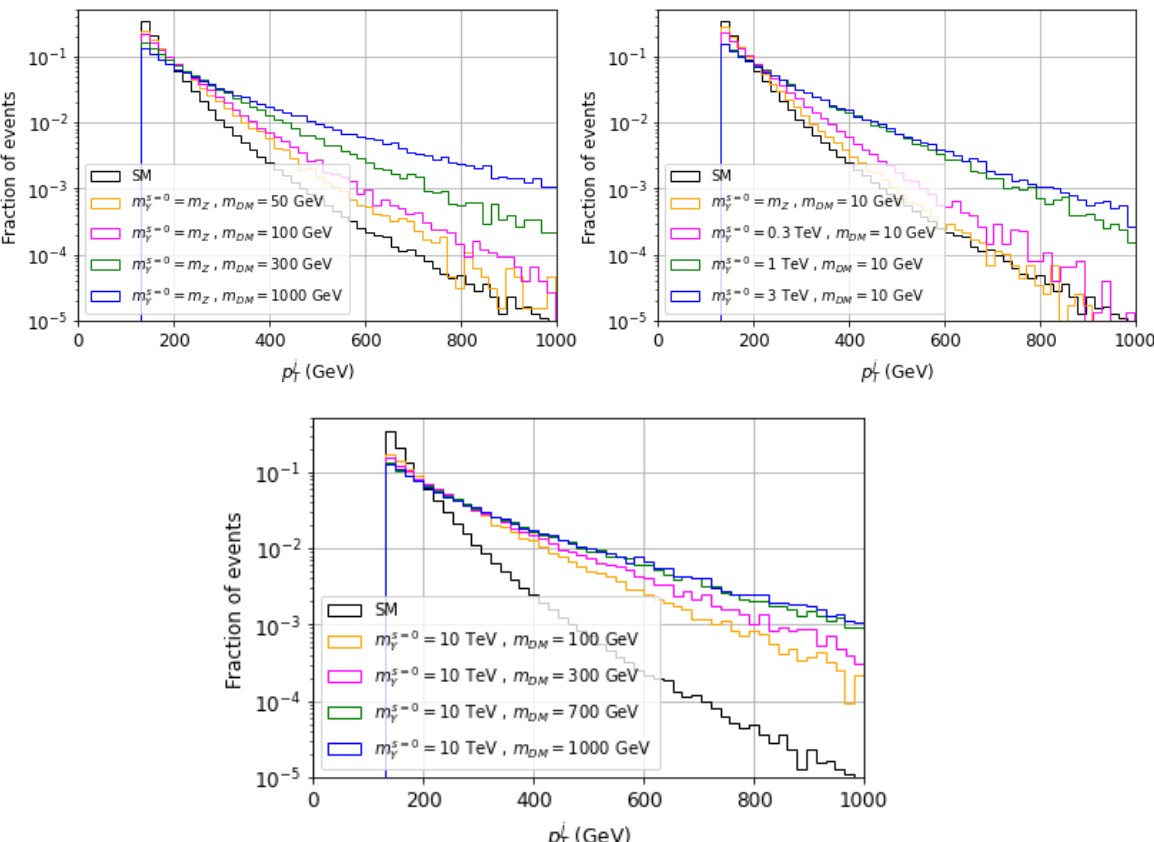

Figure 24: Space described by the $p_T^j$ distribution of the spin-0 mediator framework, *off-shell* (top-left panel), *on-shell* (top-right panel), and *off-shell PS* (bottom panel) regime, at parton level.

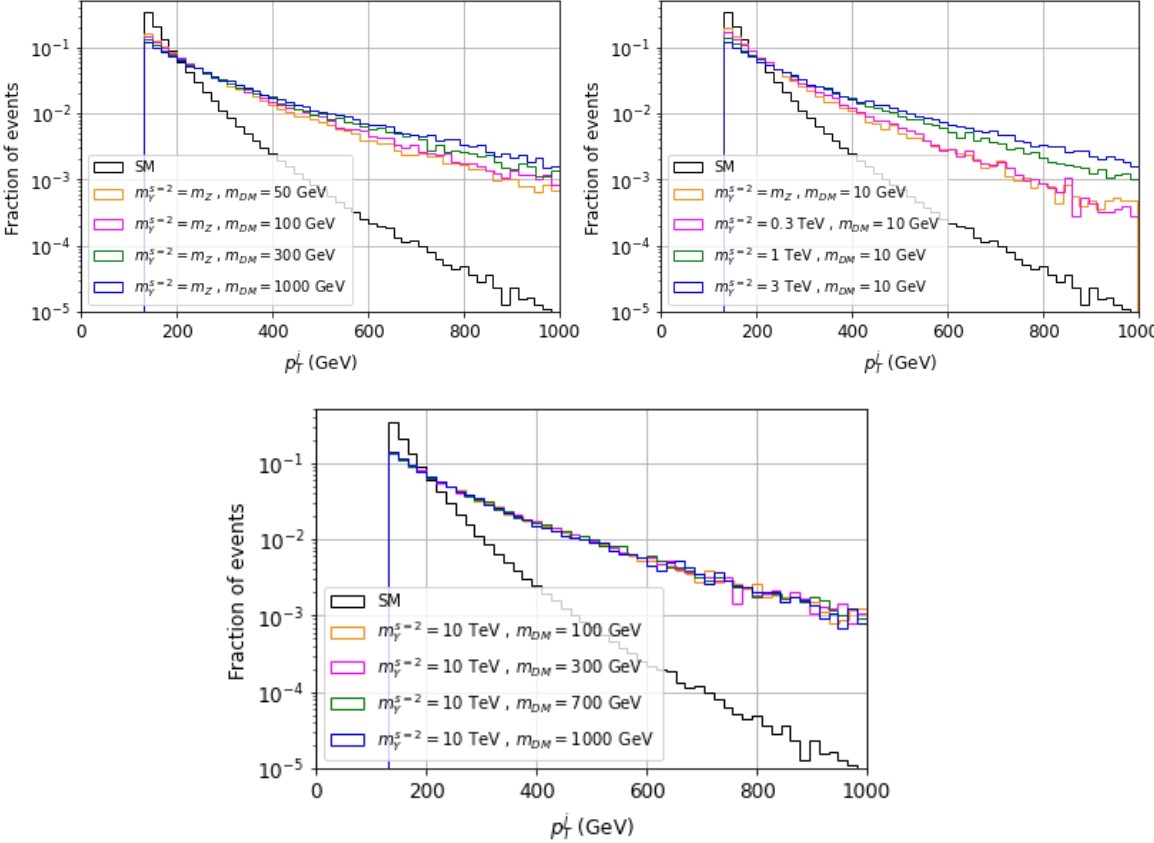

Figure 25: Space described by the $p_T^j$ distribution of the spin-2 mediator framework, *off-shell* (top-left panel), *on-shell* (top-right panel), and *off-shell PS* (bottom panel) regime, at parton level.

# D Other neural network variables

In this appendix we present more details of the neural network used. We study the dependence of the results taking parton and detector-level data as input samples. We consider two ways to define the bins for 2D histograms: dividing the phase space linearly or logarithmically. Also, we modify the total number of bins per image, and finally, we analyze the effects of different $p_T^j$ selection cut values.

## D.1 Parton vs detector-level data

Given that the kinematic distributions at parton and detector level are very similar, we expect similar performances when we apply the simulated data to a DNN. On the left panel of Fig. 26 we compare AUCs obtained with DNNs trained and tested with parton, $AUC_{parton}$, and detector-level data $AUC_{detector}$. In this and the following subsections, we employ the four models denoted *on-shell* in Section 2.3.1 as an example, but the same conclusions can be found studying the other benchmark models.

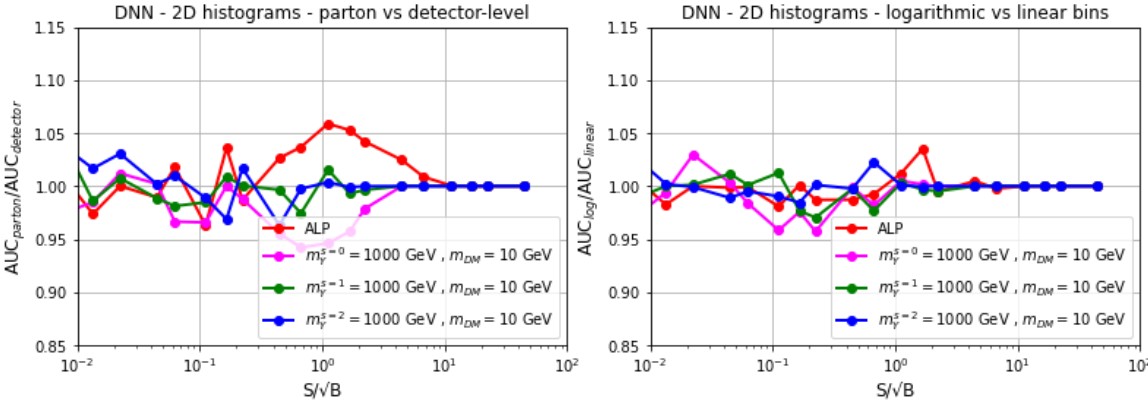

Figure 26: Left panel: DNN performance of parton versus detector-level data. Right panel: DNN performance of linear versus logarithmic representations of $p_T^j$. In both panels, the benchmark models dubbed *on-shell* are used.

## D.2 Linear vs logarithmic histograms

Throughout the main part of this work, we divided the $(\eta^j, p_T^j)$ phase space linearly to define the 2D-histogram bins. However, in Fig. 6 for example, we can notice that events (both from new physics and SM) are grouped at low $p_T^j$. Then, we can ask if a different representation with greater detail in the area of interest would result in a better performance.

On the right panel of Fig. 26 we compare the results for linear and logarithmic representations of $p_T^j$, for the benchmark models dubbed *on-shell*. No significant performance variation is found, hence we conclude that neither representation provides more information than the other.

In the next section we analyze the impact of changing the bin size, a topic related to the representation of the phase space. Increasing the overall resolution, decreasing bin size, provides more detail in $p_T^j$. Nonetheless, the DNN is sensible enough to extract sufficient information with a reasonable bin size.

## D.3 Bin size

To study the effects on the resolution, we generate 2D histograms, slicing the phase space into $N_{\text{bins}} \times N_{\text{bins}}$ bins, with $N_{\text{bins}} = 5, 10, 20, 30, 40$. We consider the benchmark models called *on-shell*. For each model, we train a DNN, whose details can be found in Table 2. The only difference between the DNNs is the number of units in the input layer, equal to the total input features, to take into account the variations in the total number of bins per histogram.

In Fig. 27 we show some examples of the constructed histograms, with different number of bins, for the SM-only sample and the spin-0 mediator model (for $N_{\text{bins}} = 30$ see Fig. 6). Naturally, we expect an increase in performance for higher resolution, but at the same time, a more complex network is needed with more parameters to train. DNN results can be seen in Fig. 28, for two examples with $S/\sqrt{B} = 2.23$ (left panel), and $S/\sqrt{B} = 0.223$ (right panel). In almost every example shown, the AUC value increases with $N_{\text{bins}}$, as expected. The most significant drop in performance is located for few number of bins, at $N_{\text{bins}} = 5$, where the histograms still can represent some of the underlying characteristics but the details are lost. Considering the behavior of all models, around $N_{\text{bins}} \sim 20 - 30$ the AUC values reach their maximum, and increasing its value at the expense of DNN complexity and computational resources for training does not seems to be justified.

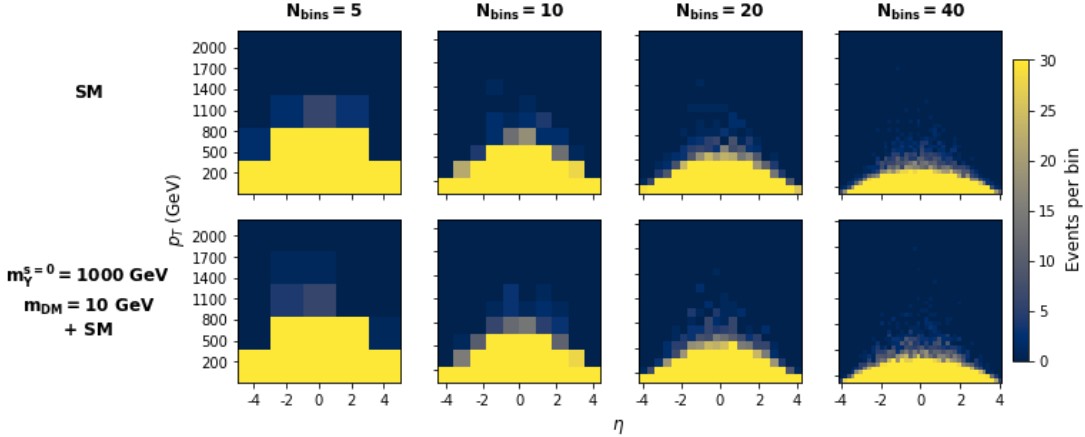

Figure 27: 2D histograms with different number of bins for the monojet final state considering SM and a spin-0 mediator model. Each figure was constructed with $B = 50k$ SM events and $S/\sqrt{B} = 2.23$ new physics events, at detector level. The color coding represents the number of events per bin, however the color scheme saturates at 30 events, to ease visualization.

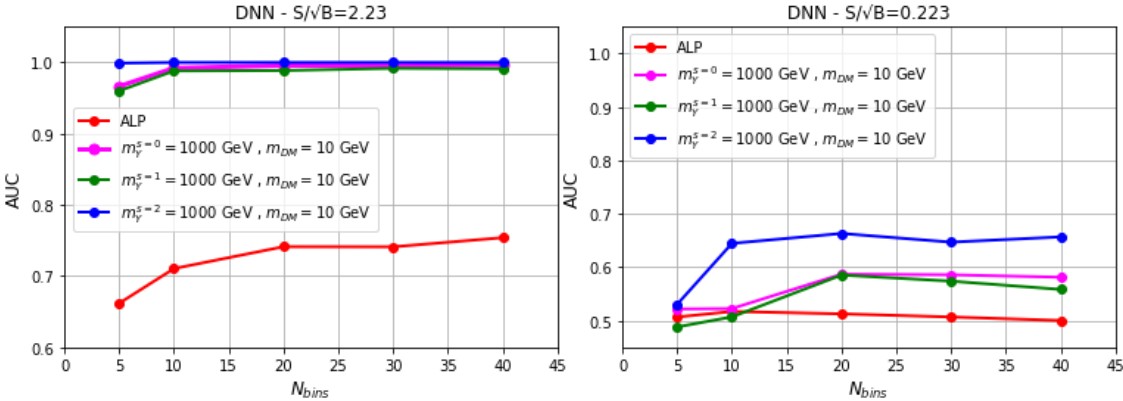

Figure 28: AUCs for histograms constructed with different number of bins, $N_{\text{bins}}$. The benchmark models dubbed *on-shell* are used, with $S/\sqrt{B} = 2.23$ (left panel), and $S/\sqrt{B} = 0.223$ (right panel).

## D.4 Minimum jet transverse momentum

In this section we analyze how the performance of DNN with histograms as input data is affected when different $p_T^j$ selection cut values are chosen. We employ the four models denoted *on-shell* as an example.

In Fig. 29 we show our results for different $p_T^j$ cuts: $130, 200, 250, 300, 350,$ and $400$ GeV. We study two signal-to-background ratios, $S/\sqrt{B} = 1.12$ and $0.223$ (left and right columns correspondingly). It is important to remark that the selected $S/\sqrt{B}$ ratios correspond to $p_T^j = 130$ GeV, for other $p_T^j$ cuts the number of signal and background events should change correspondingly. The performance as a function of $p_T^{min}$ is presented in the first row. To emulate a real situation, we assume that the collider luminosity and the couplings of each model are fixed, therefore for increasing $p_T^j$ cuts we get a decreasing amount of events. This is taken into ac-

Figure 29: AUC, $S$, and $S/\sqrt{B}$ per histogram, as a function of the selection cut $p_T^{min}$ (first, second, and third row respectively), for two examples of $S/\sqrt{B} = 1.12$ and 0.223, at $p_T^{min} = 130$ GeV (see text for details). We assume fixed collider luminosity and coupling values of each model, therefore for increasing $p_T^j$ cuts we get a decreasing amount of events.

count for the generation of the histograms to train and test the DNNs. Every set has 30×30 bins, and the number of events per histogram is shown in the second-row of the figure (to improve visualization, SM events per histogram are divided by a factor as indicated on the label). Notice that SM-background events are more suppressed than new physics ones (see also the corresponding kinematic distributions shown in the middle row of Fig. 4), therefore the $S/B$ ratio increases with $p_T^{min}$. Finally, the corresponding $S/\sqrt{B}$ values for our $p_T^{min}$ analysis are shown in the third row of Fig. 29.

In Section 4.1, with a fixed $p_T^{min}$, we showed that $S/\sqrt{B}$ is a useful parameter. From that section we expect a better (worse) DNN performance for models with increasing (decreasing) $S/\sqrt{B}$. However, comparing the first and third rows of Fig. 29 its clear that our previous conclusions do not apply when $p_T^{min}$ is not fixed. For example, the spin-1 mediator case shows for increasing $p_T^{min}$ an increasing $S/\sqrt{B}$ ratio, however the AUC decreases. In any case, AUC variations are within a few percent for the benchmark models explored.

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
