# Peer review of "Towards a method to anticipate dark matter signals with deep learning at the LHC"

_SciPost Physics, doi:SciPost Phys. 12, 063 (2022)_

## Round 2 · Referee Report · Anonymous (Referee 1) · 2021-11-21

Report

This paper presents a robust Machine Learning based method for collider dark matter searches. In particular, they focus on the monojet channel and consider several possible new physics models which could provide dark matter candidates. The problem is timely, and the method suggested is very interesting. As the authors mentioned, the idea of histograms was suggested in Ref 26, but this work thoroughly investigated the robustness of this approach. Overall, I am happy with the scientific content of the paper and believe it is suitable for publication in SciPost, but before recommending it for publication, I have the following comments:

  1. Table 2, please mention how many neurons were used in the dense layer of the CNN.

  2. I am not very sure about the statement that histograms method is totally independent of the background number of events. For example, in Fig. 10, I think there is not a significant difference because you are comparing the performance of histograms with 1000 events with 50K, 1000 is already a large number. Could you please also compare the performance of histograms with 20 events or 100 events? It seems the conclusion about the robustness against the background events is most likely true when you form a histogram of a reasonably large number of events. As mentioned in the paper, the total number of background events is decided by the luminosity, but a priori, there is no fixed number for the events to form a histogram.

  3. Section 4.3, Fig. 12, is it true that trained models should have accuracy greater than or equal to 70% because DNN trained on ALP model provides more variation?

Minor comments regarding the presentation: If possible, could you please provide better quality (pdf files) images? Though better quality images will not add scientific value, I find it distracting and difficult to read these images (see e.g. Figure 8).

I think “Machine Learning algorithm” is a more suitable name for section 3.
I also noticed a few typos in the paper. Please correct them.

---

## Round 3 · List of Changes

Re: Anonymous Report 1 on 2021-11-21 SciPost Physics

We thank the referee for her/his positive suggestions in order to improve the paper. Below we address the specific points reported:

  1. Table 2, please mention how many neurons were used in the dense layer of the CNN.

We thank again the reviewer for pointing out this issue. We have added in table 2 the missing number of neurons in the dense layer of the CNN.

  1. I am not very sure about the statement that histograms method is totally independent of the background number of events. For example, in Fig. 10, I think there is not a significant difference because you are comparing the performance of histograms with 1000 events with 50K, 1000 is already a large number. Could you please also compare the performance of histograms with 20 events or 100 events? It seems the conclusion about the robustness against the background events is most likely true when you form a histogram of a reasonably large number of events. As mentioned in the paper, the total number of background events is decided by the luminosity, but a priori, there is no fixed number for the events to form a histogram.

We have clarified this issue in the paper and we have remarked that this is true in the range of interest. We have added two panels in Fig.10 to include the cases with 20 and 100 events, and a few sentences at the end of the second paragraph of section 4.1. As expected by the referee, there is a decrease in performance for these small numbers of B, since the algorithm needs a sufficiently large number of events to represent properly the underlying distribution and perform the classification. We also included small sentences in section 6 ‘Discussion’ and in section 7 ‘Conclusions’ to specify the range of interest.

  1. Section 4.3, Fig. 12, is it true that trained models should have accuracy greater than or equal to 70% because DNN trained on ALP model provides more variation?

In Fig. 12 we are showing ratios between AUCs, so its difficult to get precise values with respect to different models. However, we can interpret the ALP results as lower limits, as we explain below.

On the top-left panel of Fig. 12 we are always using DNNs trained on ALP and testing with samples generated with the other models. Since the ALP distribution is the closest one to the SM (see Fig. 1), we obtain a significant decrease in performance. This means that the algorithm trained with ALP does not generalize correctly to be able to discriminate the other models vs SM. Notice that the biggest decrease, up to 25% (blue curve), occurs for the model with the most different distribution with respect to the SM (see Fig. 1).

In other words, if we want to choose only one model to train an algorithm and use it to distinguish background vs signal ensembles generated with different models, ALP is the worst option. In that sense, the ALP results can be interpreted as a first order lower limit.

The other panels show the results when we are using DNNs trained with the models with a mediator. We can see in Fig.12 top-right, bottom-left and bottom-right panels that these DNNs do generalize correctly and can handle test samples generated with a ‘wrong’ model.

Minor comments:

  • Figure 8 with better quality.
  • Section 3 renamed ‘Machine Learning algorithms’
  • Typos corrected.

---

## Editorial Decision

published